# Provably Efficient Exploration in Quantum Reinforcement Learning with Logarithmic Worst-Case Regret

## Abstract

While quantum reinforcement learning (RL) has attracted a surge of attention recently, its theoretical understanding is limited. In particular, it remains elusive how to design provably efficient quantum RL algorithms that can address the exploration-exploitation trade-off. To this end, we propose a novel UCRL-style algorithm that takes advantage of quantum computing for tabular Markov decision processes (MDPs) with $S$ states, $A$ actions, and horizon $H$, and establish an $\mathcal{O}(\text{poly}(S, A, H, \log T))$ worst-case regret for it, where $T$ is the number of episodes. Furthermore, we extend our results to quantum RL with linear function approximation, which is capable of handling problems with large state spaces. Specifically, we develop a quantum algorithm based on value target regression (VTR) for linear mixture MDPs with $d$-dimensional linear representation and prove that it enjoys $\mathcal{O}(\text{poly}(d, H, \log T))$ regret. Our algorithms are variants of UCRL/UCRL-VTR algorithms in classical RL, which also leverage a novel combination of lazy updating mechanisms and quantum estimation subroutines. This is the key to breaking the $\Omega(\sqrt{T})$-regret barrier in classical RL. To the best of our knowledge, this is the first work studying the online exploration in quantum RL with provable logarithmic worst-case regret.

## 1 Introduction

Reinforcement learning (RL) is ubiquitous with wide applications (Sutton & Barto, 2018). RL studies how agents take actions in an environment with the goal of maximizing cumulative reward. In light of this, a fundamental problem in RL is the balance between exploration (of uncharted territory) and exploitation (of current knowledge). This is typically characterized by the *regret* defined as the cumulative suboptimality of the policies executed by the algorithm.

One of the most notable models of RL is the Markov decision process (MDP). Among MDPs, a tabular MDP is arguably the simplest model by storing all state and action pairs in a table. Tabular MDPs have $\mathcal{O}(\text{poly}(S, A, H) \cdot \sqrt{T})$ regret (Jaksch et al., 2010; Azar et al., 2017), where $S$ and $A$ are the number of states and actions, respectively, $H$ is the length of each episode, and $T$ is the number of episodes. However, in practice $S$ and $A$ can have formidable or even infinite sizes. One typical solution is to employ linear function approximation. For a linear mixture MDP with a $d$-dimensional linear representation for the transition kernel, the regret can be improved to $\mathcal{O}(\text{poly}(d, H) \cdot \sqrt{T})$ (Ayoub et al., 2020; Cai et al., 2020; Zhou et al., 2021).

Nevertheless, for MDPs in general including tabular MDPs and linear mixture MDPs, the fundamental difficulty lies in the $\Omega(\sqrt{T})$ lower bound on their regrets (Jaksch et al., 2010; Jin et al., 2018; Zhou et al., 2021), and it solicits essentially novel ideas to break this barrier. This gives rise to *quantum computing*, which is known to achieve acceleration for some relevant problems such as search (Grover, 1997), counting (Brassard et al., 2002; Nayak & Wu, 1999), mean estimation (Montanaro, 2015; Li & Wu, 2019; Hamoudi & Magniez, 2019; Hamoudi, 2021; Cornelissen et al., 2022), etc.

In this paper, we study *quantum reinforcement learning* (Jerbi et al., 2022; Meyer et al., 2022), a combination of reinforcement learning and quantum computing, where the agent interacts with

the RL environment by accessing quantum oracles. This is in great contrast to classical reinforcement learning, where the agent interacts with the environment by sampling. The quantum oracles enable us to estimate models or value functions more efficiently than classical sampling-based algorithms, thus breaking the $\Omega(\sqrt{T})$ regret lower bound. For example, Wan et al. (2022) showed that logarithmic regret is possible in quantum bandits, a simpler task than quantum reinforcement learning. Currently, whether quantum algorithms can efficiently solve MDPs remains elusive. The only exception is Wang et al. (2021a), which developed fast quantum algorithms for solving tabular MDPs. However, their work requires a generative model (Kakade, 2003), thus it did not address the exploration-exploitation trade-off, a core challenge in online RL. In this work, we focus on online quantum RL and aim to answer the following question:

*Can we design algorithms that enjoy logarithmic worst-case regret for online quantum RL?*

**Contributions.** We study the exploration problem in quantum reinforcement learning and establish $\mathrm{poly}(\log T)$ regret bound, where $T$ is the number of episodes. In specific,

- For tabular MDPs, we propose a novel algorithm Quantum UCRL (Algorithm 4) and prove the worst-case regret guarantee of $\mathcal{O}(\mathrm{poly}(S, A, H, \log T))$.
- For linear mixture MDPs where the transition kernel permits a $d$-dimensional linear representation, we develop the algorithm Quantum UCRL-VTR (Algorithm 1) with worst-case $\mathcal{O}(\mathrm{poly}(d, H, \log T))$ regret guarantee.

Hence, we break the $\Omega(\sqrt{T})$-regret barrier in classical RL as desired. To our best knowledge, we present *the first line of study on exploration in online quantum RL*.

**Challenges and Technical Overview.** Classical RL lower bound shows that $\sqrt{T}$ regret is inevitable and we need to use quantum tools to achieve speedup. Our key observation is that quantum computing can achieve speedup in mean estimation problems (see Lemma 3.1 and Lemma 3.2). Intuitively, as quantum mean estimation provides more accurate estimations, it allows the agent to engage in more conservative exploration, resulting in a more refined regret bound. There are three main challenges to adapting the quantum mean estimation subroutines in RL problems: **(1)** The first challenge is how to collect data for the quantum mean estimation subroutines to estimate the model. In quantum RL, the observation is a sequence of quantum states. However, upon measuring these quantum states, their collapse ensues, avoiding their utilization in the model estimation. Conversely, if we abstain from measuring these quantum states, we face the challenge of how to design strategic exploration policy. **(2)** The second challenge is the difficulty of reusing the quantum samples. As we have explained above, the quantum samples will collapse after each estimation, which means that the desired quantum algorithms cannot update model estimation in each episode similar to the classical UCRL and UCRL-VTR algorithms. **(3)** Classical linear RL analysis heavily relies on martingale concentration (Ayoub et al., 2020). However, in quantum RL, there is no direct counterpart to martingale analysis.

We propose the following novel techniques to resolve the above challenges, more details and technical comparisons with existing works are deferred to Appendix B.2. **(1)** For the first challenge, we observe that the quantum oracles are able to imitate the classical sampling at a cost of an additional $H$ term in the regret (see the CSQA subroutine in Appendix C.2). Thus, we can collect quantum samples by sampling a classical $(s, a)$ first, and then querying the transition oracle on $|s, a\rangle$ to obtain a quantum sample used to estimate the model. **(2)** To mitigate the second challenge, we use the doubling trick to design lazy-updating algorithms, which address the challenge of reusing quantum samples by estimating the model less frequently while still achieving efficient quantum speedup. Specifically, for tabular MDPs, we propose the Quantum UCRL algorithm, a variant of UCRL2 (Jaksch et al., 2010), which utilizes the doubling trick to the visiting times of each state-action pair to ensure the lazy updating and adopts the quantum multi-dimensional amplitude estimation (Lemma 3.1) to achieve quantum speedup. For linear mixture MDPs, we develop the Quantum UCRL-VTR algorithm (Algorithm 1), which is a variant of UCRL-VTR (Ayoub et al., 2020). The Quantum UCRL-VTR algorithm performs the doubling trick in the determinant of the covariance matrix to ensure the lazy updating property. This algorithm design, absent in previous work on quantum RL with a generative model (Wang et al., 2021b), connects online exploration in quantum RL with strategic lazy updating mechanisms. **(3)** we redesign the lazy updating frequency and propose an entirely new technique to estimate features (Algorithm 2) for building the confidence set for the

model. Our algorithm design and analysis successfully handle some subtle technical issues like estimating an unknown feature in each episode up to some accuracy that depends on the estimated feature itself (cf. (5.5)). To the best of our knowledge, this algorithm design (Algorithm 2) and theoretical analysis are entirely novel in the literature on (classical and quantum) RL theory. See "feature estimation" part of Section 5 and Appendix B.2 for more details.

**Related Works.** Broadly speaking, our work is related to quantum machine learning, we will discuss related works in Appendix A due to space limit. Among previous works on quantum machine learning, our work is mostly related to Wang et al. (2020) on quantum RL with a generative model. We have emphasized that we focus on the more challenging online quantum RL. Our work is also related to previous works on classical online RL (e.g., Jaksch et al., 2010; Azar et al., 2017; Jin et al., 2018; Ayoub et al., 2020), where the regret lower bound is $\Omega(\sqrt{T})$. In comparison, our quantum RL algorithms enjoy logarithmic regret. See Appendix A for more discussions on classical RL.

## 2 PRELIMINARIES

**Episodic Markov Decision Processes.** An episodic finite-horizon Markov decision process (MDP) can be described by a tuple $\mathcal{M} = (\mathcal{S}, \mathcal{A}, H, \mathcal{P} = \{\mathcal{P}_h\}_{h\in[H]}, R = \{R_h\}_{h\in[H]})$, where $\mathcal{S}$ is the state space, $\mathcal{A}$ is the action space, $\mathcal{P}_h$ is the transition kernel at step $h$, $R_h$ is the reward function at the $h$-th step. For simplicity, we assume the reward function is known and deterministic. This is a reasonable assumption since learning transition kernels is more challenging than learning reward functions and our subsequent results are ready to be extended to the unknown stochastic reward functions.

At the beginning of each episode, the agent determines a policy $\pi = \{\pi_h : \mathcal{S} \mapsto \Delta(\mathcal{A})\}_{h\in[H]}$. Without loss of generality, we assume that each episode starts from a fixed state $s_1 \in \mathcal{S}^1$. At each step $h$, the agent receives a state $s_h$, takes an action $a_h \sim \pi_h(\cdot \,|\, s_h)$, receives the reward $r_h = R_h(s_h, a_h)$, and transits to the next state $s_{h+1} \sim \mathcal{P}_h(\cdot \,|\, s_h, a_h)$. The episode ends after receiving $r_H$.

Given a policy $\pi$, its value function $V_h^\pi \colon \mathcal{S} \mapsto \mathbb{R}$ at the $h$-th step is defined as $V_h^\pi(s) = \mathbb{E}_\pi[\sum_{h'=h}^H R_{h'}(s_h, a_h) \,|\, s_h = s]$, where the expectation $\mathbb{E}_\pi$ is taken with respect to the randomness induced by the transition kernels and policy $\pi$. Correspondingly, given a policy $\pi$, we define its $h$-th step Q-function $Q_h^\pi \colon \mathcal{S} \times \mathcal{A} \mapsto \mathbb{R}$ as $Q_h^\pi(s, a) = \mathbb{E}_\pi[\sum_{h'=h}^H R_{h'}(s_h, a_h) \,|\, s_h = s, a_h = a]$. It is well known that the value function and the Q-function satisfy the following Bellman equation:

$$Q_h^\pi(s, a) = R_h(s, a) + [\mathbb{P}_h V_{h+1}^\pi](s, a), \ V_h^\pi(s) = \langle Q_h^\pi(s, \cdot), \pi_h(\cdot \,|\, s)\rangle_\mathcal{A}, \ V_{H+1}^\pi(s) = 0, \quad (2.1)$$

for any $(s, a) \in \mathcal{S} \times \mathcal{A}$. Here, $\mathbb{P}_h$ is the operator defined as

$$(\mathbb{P}_h V)(s, a) = \mathbb{E}\left[V(s') \,|\, s' \sim \mathcal{P}_h(\cdot \,|\, s, a)\right] \quad (2.2)$$

for any function $V \colon \mathcal{S} \mapsto \mathbb{R}$. The online exploration problem in reinforcement learning requires learning the optimal policy $\pi^*$ by interacting with the episodic MDP, which by definition maximizes the value function, i.e., $V_h^{\pi^*}(s) = \max_\pi V_h^\pi(s)$ for all $s \in \mathcal{S}$ and $h \in [H]$. To simplify the notation, we use the shorthands $V_h^* = V_h^{\pi^*}$ and $Q_h^* = Q_h^{\pi^*}$ for all $h \in [H]$. We also use $d_h^\pi(\cdot) := \Pr_\pi(s_h = \cdot)$ to denote the occupancy measure over the states.

We use the notion of regret to measure the performance of the agent. Suppose policy $\pi^t$ is executed in the $t$-th episode for $t \in [T]$, then the regret for $T$ episodes is defined as $\text{Regret}(T) = \sum_{t=1}^T [V_1^*(s_1) - V_1^{\pi^t}(s_1)]$. We aim to design algorithms minimizing the regret.

**Linear Function Approximation.** To tackle the large state space, we also consider the function approximation. In this paper, we focus on the linear mixture MDP (Ayoub et al., 2020; Modi et al., 2020; Cai et al., 2020), where the transition kernel is linear in a feature map.

**Definition 2.1** (Linear Mixture MDP). We say an MDP $(\mathcal{S}, \mathcal{A}, H, \mathcal{P}, R)$ is a linear mixture MDP if there exists a known feature map $\psi \colon \mathcal{S} \times \mathcal{A} \times \mathcal{S} \to \mathbb{R}^d$ and unknown vectors $\{\theta_h \in \mathbb{R}^d\}_{h\in[H]}$ with $\|\theta_h\|_2 \le 1$ such that

$$\mathcal{P}_h(s' \,|\, s, a) = \psi(s, a, s')^\top \theta_h \quad (2.3)$$

---

[1]Our subsequent results can be easily extended to the case where the initial state is sampled from a fixed distribution.

for all $(s, a, s', h) \in \mathcal{S} \times \mathcal{A} \times \mathcal{S} \times [H]$. Moreover, for any $(s, a) \in \mathcal{S} \times \mathcal{A}$ and $V : \mathcal{S} \mapsto [0, 1]$, we assume that

$$\|\phi_V(s, a)\|_2 \leq 1, \tag{2.4}$$

where we use the notation that

$$\phi_V(s, a) = \int_{s'} \psi(s, a, s') \cdot V(s') \mathrm{d}s'. \tag{2.5}$$

When $d = |\mathcal{S}|^2|\mathcal{A}|$ and the feature map $\psi(s, a, s')$ being the canonical basis $\mathbf{1}_{s,a,s'} \in \mathbb{R}^{|\mathcal{S}|^2|\mathcal{A}|}$, linear mixture MDPs reduce to the tabular MDP. Also, linear mixture MDPs include another linear type MDPs proposed by Yang & Wang (2020) as a special case. Finally, we remark that Yang & Wang (2019); Jin et al. (2020) study the linear MDP, which is different from the linear mixture MDP. Linear mixture MDPs and linear MDPs are incomparable in the sense that one cannot include the other as the special case.

## 3 QUANTUM REINFORCEMENT LEARNING

In this section, we introduce basic concepts about quantum computing and quantum-accessible reinforcement learning. Further explanations are also given in Appendix C.

### 3.1 QUANTUM COMPUTING

**Basics.** Consider a classical system that has $d$ different states, indexed from 0 to $d-1$. We call these states $|0\rangle, \ldots, |d-1\rangle$, where $|i\rangle$ is identical to the classical state $i$. A *quantum state* $|v\rangle$ is a *superposition* of the classical states

$$|v\rangle = \sum_{i=0}^{d-1} v_i |i\rangle \tag{3.1}$$

where $v_i \in \mathbb{C}$ is called the *amplitude* of $|i\rangle$, $|v\rangle$ is normalized and $\sum_{i=0}^{d-1} |v_i|^2 = 1$.

Mathematically, $|0\rangle, \ldots, |d-1\rangle$ forms an orthonormal basis of a $d$-dimensional Hilbert space. As an example, a classical state $s \in \mathcal{S}$ is related to a quantum state $|s\rangle \in \bar{\mathcal{S}}$ in quantum reinforcement learning, where $\bar{\mathcal{S}}$ is a large Hilbert space containing all the superposition of classical states. We call $\{|s\rangle\}_{s \in \mathcal{S}}$ as the computational basis of $\bar{\mathcal{S}}$. Quantum states from different Hilbert spaces can be combined with tensor product. For notation simplicity, we use $|v\rangle|w\rangle$ or $|v, w\rangle$ to denote the tensor product $|v\rangle \otimes |w\rangle$. Operations in quantum computing are *unitaries*, i.e., a linear transformation $U$ such that $U^\dagger U = UU^\dagger = I$ ($U^\dagger$ is the conjugate transpose of $U$).

**Information transfer between quantum and classical computers.** The information in a quantum state cannot be "seen" directly. Instead, to observe a quantum state $|v\rangle$, we need to perform a *quantum measurement* on it. The measurement gives a classical state $i$ with probability $|v_i|^2$, and the measured quantum state becomes $|i\rangle$, losing all its information. As an example, a measurement for quantum state $|i\rangle$ returns $i$ with probability exactly 1.

Quantum access to the input data is encoded in a unitary operator called *quantum oracle*. There are different common input models in quantum computing, such as the probability oracle (Definition C.1) and the binary oracle (Definition C.2). By quantum oracles the input data can be read in *superposition*, i.e., they allow us to efficiently obtain a quantum state $|\phi\rangle = \sum_{s \in \mathcal{S}} a_s |s\rangle$ defined in (3.1), and perform operations on these $|s\rangle$ "in parallel", which is the essence of quantum speedups.

In our algorithm, we consider the *query complexity* as the number of queries to the quantum oracles.

**Quantum multi-dimensional amplitude estimation and multivariate mean estimation.** To achieve quantum speedup, we exploit two standard quantum subroutines: the quantum multi-dimensional amplitude estimation (van Apeldoorn, 2021) and quantum multivariate mean estimation (Cornelissen et al., 2022) stated below.

**Lemma 3.1** (Quantum multi-dimensional amplitude estimation, Rephrased from Theorem 5 of van Apeldoorn 2021)**.** Assume that we have access to the probability oracle $U_p \colon |0\rangle \rightarrow$

$\sum_{i=0}^{n-1} \sqrt{p_i}|i\rangle|\phi_i\rangle$ for an $n$-dimensional probability distribution $p$ and ancilla quantum states[2] $\{|\phi_i\rangle\}_{i=0}^{n-1}$. Given parameters $\varepsilon > 0$, $\delta > 0$, an approximation $\widetilde{p}$ such that $||p - \widetilde{p}||_1 \le \varepsilon$ can be found with probability $\ge 1 - \delta$ using $\mathcal{O}(n \log(n/\delta)/\varepsilon)$ quantum queries to $U_p$ and its inverse.

**Lemma 3.2** (Quantum multivariate mean estimation, Rephrased from Theorem 3.3 of Cornelissen et al. 2022). Let $X: \Omega \to \mathbb{R}^d$ be a $d$-dimensional bounded variable on probability space $(\Omega, p)$ such that $||X||_2 \le C$ for some constant $C$. Assume that we have access to (i) the probability oracle $U_p: |0\rangle \to \sum_{\omega \in \Omega} \sqrt{p(\omega)}|\omega\rangle|\phi_\omega\rangle$ for ancilla quantum states $\{|\phi_\omega\rangle\}_{\omega \in \Omega}$; and (ii) the binary oracle $U_X: |\omega\rangle|0\rangle \to |\omega\rangle|X(\omega)\rangle$, $\forall \omega \in \Omega$. Given two reals $\delta \in (0, 1)$ and $\varepsilon > 0$, there exists a quantum algorithm outputs a mean estimate $\widetilde{\mu}$ of $\mu = \mathbb{E}[X]$ such that $||\widetilde{\mu}||_2 \le C$ and $||\widetilde{\mu} - \mu||_2 \le \varepsilon$ with probability $\ge 1 - \delta$, using $\mathcal{O}(C\sqrt{d} \log(d/\delta)/\varepsilon)$ quantum queries to $U_p$, $U_X$, and their inverses.

## 3.2 QUANTUM-ACCESSIBLE ENVIRONMENTS

In this paper, we hope to study the online exploration problem in quantum reinforcement learning by leveraging powerful tools in quantum computing. To this end, we introduce the quantum-accessible RL environments in this section.

Suppose we have two Hilbert spaces $\bar{\mathcal{S}}$ and $\bar{\mathcal{A}}$ that contain the superpositions of the classical states and actions. We use $\{|s\rangle\}_{s \in \mathcal{S}}$ and $\{|a\rangle\}_{a \in \mathcal{A}}$ as the computational basis in $\bar{\mathcal{S}}$ and $\bar{\mathcal{A}}$, respectively. Following the quantum-accessible environments studied by previous works (Wang et al., 2021a; Jerbi et al., 2022; Wiedemann et al., 2022), we use two quantum oracles to access the episodic MDP $\mathcal{M}$ for each step $h \in [H]$:

- The transition oracle $\bar{\mathcal{P}} = \{\bar{\mathcal{P}}_h\}_{h=1}^H$ that returns the superposition over next states according to the transition probability $\mathcal{P}_h$, which is a quantum probability oracle.

$$\bar{\mathcal{P}}_h : |s_h, a_h\rangle|0\rangle \to |s_h, a_h\rangle \otimes \sum_{s_{h+1}} \sqrt{\mathcal{P}_h(s_{h+1} \mid s_h, a_h)}|s_{h+1}\rangle. \tag{3.2}$$

- The reward oracle $\bar{\mathcal{R}} = \{\bar{\mathcal{R}}_h\}_{h=1}^H$ that returns the binary representation of the reward.[3]

$$\bar{\mathcal{R}}_h : |s_h, a_h\rangle|0\rangle \to |s_h, a_h\rangle|R_h(s_h, a_h)\rangle. \tag{3.3}$$

As long as a classical RL task can be written as a computer program with source code, we can perform our quantum RL algorithm with these quantum oracles. This is because such classical programs can in principle be written as a Boolean circuit whose output follows the distribution $s_{h+1} \sim \mathcal{P}_h(\cdot \mid s_h, a_h)$, and it is known that any classical circuit with $N$ logic gates can be converted to a quantum circuit consisting of $O(N)$ logic gates that can compute on any quantum superposition of inputs (see for instance Section 1.5.1 of Nielsen & Chuang (2002) and Wang et al. (2021a)), which gives the quantum state $\sqrt{\mathcal{P}_h(s_{h+1} \mid s_h, a_h)}|s_{h+1}\rangle$ in (3.2). As an intuitive example, Atari games can be played with our quantum oracles when quantum computers become universal and have enough number of qubits to execute the programs. In all, we assume the ability to call these quantum oracles and their inverse.

In classical RL environments, the agent interacts with any MDP $\mathcal{M}$ by determining a stochastic policy $\pi = \{\pi_h : \mathcal{S} \mapsto \Delta(\mathcal{A})\}_{h \in [H]}$. Analogously, we introduce quantum-evaluation of a policy in quantum-accessible RL environments (Wiedemann et al., 2022; Jerbi et al., 2022). The quantum evaluation of a classical policy $\pi$ is $H$ unitaries $\Pi = \{\Pi_h\}_{h=1}^H$ such that

$$\Pi_h : |s\rangle|0\rangle \to |s\rangle \sum_a \sqrt{\pi_h(a \mid s)}|a\rangle \quad \forall h \in [H]. \tag{3.4}$$

The unitary $\Pi_h$ quantizes the randomness of $\pi_h$ into the quantum state $\sqrt{\pi_h(a \mid s)}|a\rangle$. Any policy that is classically computable can be converted to such unitaries in quantum computation efficiently (Grover & Rudolph, 2002; Jerbi et al., 2022).

---

[2]Ancilla quantum states help and broaden the scope of quantum computing tasks. The simplest case is that all states $|\phi_i\rangle$ are identical and we can remove this state. In general, Lemma 3.1 and Lemma 3.2 hold for any such oracle $U_p$.

[3]Note that we have assumed that the reward $R$ is known for simplicity. It is straightforward to extend to the unknown reward setting with quantum access to $\bar{\mathcal{R}}$ (Auer et al., 2008).

Quantum probability oracles are more powerful than classical sampling in the sense that we can simulate classical sampling of $s_h \sim d_h^\pi$ by quantum oracles using the Classical Sampling via Quantum Access (CSQA) subroutine (Algorithm 3) for an input policy $\pi$ and target step $h$. The CSQA subroutine computes a quantum state $\varphi_h = \sum_s \sqrt{d_h^\pi(s)}|s\rangle$ using one call to $\bar{\mathcal{P}}_{h'}$ and $\Pi_{h'}$ for each $h' < h$ (the classical sampling requires one sample from $\mathcal{P}_{h'}$ and $\pi_{h'}$ correspondingly). Therefore, it suffices to measure $\varphi_h$ to obtain a classical sample $s_h \sim d_h^\pi$.

### 3.3 Quantum Exploration Problem

In the classical exploration problem, the agent interacts with the environment by executing policy $\pi_h$ to take an action $a_h \sim \pi_h(\cdot \mid s_h)$ based on the current state $s_h$, and then transiting to $s_{h+1} \sim \mathcal{P}_h(\cdot \mid s_h, a_h)$ for each step $h$ in an episode. In this paper, we study the exploration problem in quantum-accessible environments (also called the quantum exploration problem), which is a natural extension of the classical exploration problem.

In the quantum exploration problem, the agent "executes" a policy $\pi_h$ by calling the quantum evaluation $\Pi_h$ of $\pi_h$, and then "transits" to the next state by calling the transition oracle $\bar{\mathcal{P}}_h$. This correspondence is natural in that $\Pi_h$ and $\bar{\mathcal{P}}_h$ exactly quantize the randomness of $\pi_h$ and $\mathcal{P}_h$. The oracles $\Pi_h$ and $\bar{\mathcal{P}}_h$ are allowed to be called once in an episode.

Prior to this paper, Wang et al. (2021a) studied the learning of optimal policies on discounted MDP under quantum-accessible environments. However, they assumed the ability to prepare any quantum state $|s\rangle$ for all $s \in \mathcal{S}$, which enables them to use $\bar{\mathcal{P}}$ and $\bar{\mathcal{R}}$ as generative models to query any state and action (Sidford et al., 2018; Kearns & Singh, 1998). In online RL, however, the agent cannot access unexplored states. As a consequence, the agent has to actively explore the unknown environment to learn the high-rewarded area of the state space. This is known as the exploration challenge, which is ubiquitous in the literature of online RL (Auer et al., 2008; Azar et al., 2017; Jin et al., 2018). In the quantum exploration problem, we cannot prepare arbitrary $|s\rangle$, and we need to resolve the exploration challenge.

## 4 Warmup: Results for Tabular MDPs

As an initial study of online quantum RL, we focus on the tabular setting where the state and action space are finite and of small sizes, where we assume $|\mathcal{S}| = S, |\mathcal{A}| = A$. For tabular RL, we propose Quantum UCRL (Algorithm 4 in Appendix E) that learns the optimal policy given quantum access to an episodic MDP. Its key ingredients are summarized as follows.

**Data collection scheme.** Unlike classical RL that a complete trajectory $(s_1, a_1, s_2, a_2, ..., s_H, a_H)$ is revealed in a single episode, we can only collect quantum states that cannot be seen directly in quantum-accessible environments. Therefore, we cannot build the estimators of $\mathcal{P}_h(\cdot \mid s_h, a_h)$ with $s_{h+1}$ as in classical RL algorithms (Auer et al., 2008; Azar et al., 2017) because $s_h, a_h, s_{h+1}$ are not directly observable without measurements.

To this end, we divide $T$ episodes into different phases, where one phase consists of $H$ consecutive episodes. We use a fixed policy $\pi^k$ during the $k$-th phase to collect quantum samples for each $h \in [H]$. To construct an estimator of $\mathcal{P}_h(\cdot \mid s_h, a_h)$ for any $(s_h, a_h)$, we need to first obtain a classical sample $(s_h, a_h)$, then query $\bar{\mathcal{P}}_h$ on $|s_h, a_h\rangle$ to get a quantum sample of $\mathcal{P}_h(\cdot \mid s_h, a_h)$. Quantum subroutines enable us to estimate $\mathcal{P}_h(\cdot \mid s_h, a_h)$ with quantum samples more efficiently. Fortunately, we can accomplish this classical sampling by the CSQA subroutine, and query $\bar{\mathcal{P}}_h$ once to acquire a quantum sample.

**Lazy updating via doubling trick.** Fix a state-action pair $(s, a)$ and step $h$, it requires $\widetilde{\mathcal{O}}(S/\epsilon)$ quantum samples of $\mathcal{P}_h(\cdot \mid s, a)$ (i.e., $\widetilde{\mathcal{O}}(S/\epsilon)$ calls to $\bar{\mathcal{P}}_h$ or its inverse on $|s, a\rangle$) to form an $\epsilon$-close estimator in terms of $\ell_1$ distance (Lemma 3.1). However, the quantum estimation subroutine requires to do a measurement in the end, which causes all the quantum samples to collapse. As a result, these quantum samples are not reusable in the future estimation, in contrast to the classical setting where each sample can be reused to construct future estimators.

This phenomenon has been observed by Wan et al. (2022) in quantum bandits, where they also need to design a lazy updating scheme to estimate the rewards of each arm. In the quantum bandit

problems, the agent is able to constantly pull an arm and collect quantum samples to estimate it, while it is not possible to constantly query on the same $(s, a)$ in MDPs. Therefore, we have to design a more complicated lazy updating scheme.

We use a doubling trick to resolve this issue. Define the tag function by $l_h(\cdot, \cdot) : \mathcal{S} \times \mathcal{A} \to \mathbb{N}$, which are initially 0. We only reconstruct the estimator $\widehat{\mathcal{P}}_h(\cdot \mid s, a)$ using quantum multi-dimensional amplitude estimation (Lemma 3.1) as long as the number of quantum samples $n_h(s, a)$ reaches $2^{l_h(s,a)}$. Then we add $l_h(\cdot, \cdot)$ by 1 to double the length of this procedure. In this way, we ensure $n_h(s, a)/2 \leq \widetilde{n}_h(s, a) \leq n_h(s, a)$, where $\widetilde{n}_h(s, a)$ is the number of quantum samples actually used to build the estimator. More details on the counter updating is deferred to Remark C.4.

**Optimistic planning.** Thanks to the quantum subroutines and doubling trick, we can construct an estimator $\widehat{\mathcal{P}}_h(\cdot \mid s, a)$ for any $(s, a)$ such that with high probability

$$\left\| \widehat{\mathcal{P}}_h(\cdot \mid s, a) - \mathcal{P}_h(\cdot \mid s, a) \right\|_1 \leq \widetilde{\mathcal{O}}\left( \frac{S}{n_h(s, a)} \right). \tag{4.1}$$

This greatly improves the $\widetilde{\mathcal{O}}(1/\sqrt{n_h(s, a)})$ rate of empirical estimation in classical RL (Jaksch et al., 2010). See Appendix D.1 for the detailed implementation.

At the end of each phase, we update the optimistic value functions by optimistic value iteration (Azar et al., 2017) with bonus function $b_h(s, a) := \widetilde{\mathcal{O}}(HS/n_h(s, a))$ (see the formal definition in (E.1)):

$$Q_h(s, a) = \min\{R_h(s, a) + \widehat{\mathbb{P}}_h V_{h+1}(s, a) + b_h(s, a), H\}, \tag{4.2}$$

$$V_h(s) = \max_a Q_h(s, a), \tag{4.3}$$

where $\widehat{\mathbb{P}}_h$ is the operator defined by $(\widehat{\mathbb{P}}_h V)(s, a) = \mathbb{E}[V(s') \mid s' \sim \widehat{\mathcal{P}}_h(\cdot \mid s, a)]$. The exploration policy $\pi^{k+1}$ for the next phase is fixed as the greedy policy with regards to $Q$.

The theoretical guarantee of Algorithm 4 is given below.

**Theorem 4.1.** With probability at least $1 - \delta$, the regret of Quantum UCRL (Algorithm 4) is at most

$$\text{Regret}(T) = \mathcal{O}\left( S^2 A H^3 \log(T) \log(S^2 A H \log(T)/\delta) \right).$$

The detailed proof of Theorem 4.1 is deferred to Appendix E. Our results show that it is possible to design novel algorithms with only $\mathcal{O}(\text{poly}(\log T))$ regret with the help of quantum subroutines. This greatly improves on the classical setting where the regret bound of any classical algorithm must be at least $\Omega(\sqrt{T})$ (Auer et al., 2008; Jin et al., 2018). We conjecture that the dependence of $S$ and $H$ in Theorem 4.1 can be improved. Specifically, to improve the dependency on $S$, it may be necessary to design provably efficient model-free algorithms since the regret $\mathcal{O}(S^2 A \log(T))$ seems inevitable for model-based algorithms (Azar et al., 2017; Zhang et al., 2021). On the other hand, to improve the dependency on $H$, one may need to utilize the technique of variance reduction (Azar et al., 2017).

## 5  RESULTS FOR LINEAR MIXTURE MDPS

The result for tabular MDPs is a simple demonstration of the effectiveness of quantum RL. Our main setting is quantum RL with linear function approximation. In specific, we customize a novel quantum RL algorithm for linear mixture MDP (Modi et al., 2020; Ayoub et al., 2020; Cai et al., 2020), followed by theoretical guarantees.

We present the Quantum UCRL-VTR algorithm in Algorithm 1. The learning process consists of $T$ episodes, which are divided into $K$ phases. At each phase, the algorithm has four ingredients: (i) model estimation; (ii) optimistic planning; (iii) feature estimation; and (iv) regression targets estimation. We present the details of the $k$-th phase below.

**Model estimation.** At the beginning of the $k$-th phase, the agent has access to the estimators obtained in previous $k - 1$ phases, including

- Estimated value functions $\{V_h^\tau\}_{(\tau, h) \in [k-1] \times [H]}$;
- Estimated features $\{\widehat{\phi}_h^\tau\}_{(\tau, h) \in [k-1] \times [H]}$, which are estimators of

$$\left\{ \phi_h^\tau := \mathbb{E}_{(s_h, a_h) \sim \pi^\tau}[\phi_{V_{h+1}^\tau}(s_h, a_h)] \right\}_{(\tau, h) \in [k-1] \times [H]}.$$

---

**Algorithm 1** Quantum UCRL-VTR

---
1: **for** phase $k = 1, \cdots, K$ **do**
2:      Calculate $\{\bar{\theta}_h^k\}_{h \in [H]}$ and $\{\Lambda_h^k\}_{h \in [H]}$ as (5.3).
3:      Construct the confidence set $\mathcal{C}^k = \{\mathcal{C}_h^k\}_{h \in [H]}$ as (5.4).
4:      $(\{Q_h^k\}_{h \in [H]}, \{V_h^k\}_{h \in [H]}, \{\pi_h^k\}_{h \in [H]}) \leftarrow$ Optimistic Planning($\mathcal{C}^k$) (Algorithm 5).
5:      $\{\widehat{\phi}_h^k\}_{h \in [H]} \leftarrow$ Estimate Feature($\{\Lambda_h^k\}_{h \in [H]}$) (Algorithm 2).
6:      Set $w_k \leftarrow \max_{h \in [H]} \|\widehat{\phi}_h^k\|_{(\Lambda_h^k)^{-1}}$.
7:      $\{y_h^k\}_{h \in [H]} \leftarrow$ Estimate Regression Target($\pi^k, V^k, w_k$) (Algorithm 6).
8: **end for**

---

**Algorithm 2** Estimate Feature

---
1: **Input:** Positive definite matrices $\{\Lambda_h\}_{h \in [H]}$ .
2: Initialize: $m = 1$, $\widehat{\phi}_{h,0} = \widehat{\phi}_{h,1} = \mathbf{0}, \forall h \in [H]$.
3: **while** $\|\widehat{\phi}_{h,m-1}\|_{(\Lambda_h)^{-1}} < 2^{2-m}$ for all $h \in [H]$ **do**
4:      $m \leftarrow m + 1$.
5:      **for** $h = 1, \cdots, H$ **do**
6:          Calculate $\widehat{\phi}_{h,m}$ by quantum multivariate mean estimation subroutine (Lemma 3.2) up to error $2^{-m}$.
7:      **end for**
8: **end while**

---

- Regression targets $\{y_h^\tau\}_{(\tau,h) \in [k-1] \times [H]}$, which are estimators of

$$\left\{\mathbb{E}_{(s_h,a_h) \sim \pi^\tau}[\mathbb{P}_h V_{h+1}^\tau(s_h, a_h)]\right\}_{(\tau,h) \in [k-1] \times [H]}.$$

Note that, for any $h \in [H]$,

$$\mathbb{E}_{\pi^\tau}[\mathbb{P}_h V_{h+1}^\tau(s_h, a_h)] = \mathbb{E}_{\pi^\tau}\left[\int_{s'} V_{h+1}^\tau \mathcal{P}_h(s' \mid s_h, a_h)\mathrm{d}s'\right]$$
$$= \mathbb{E}_{\pi^\tau}\left[\int_{s'} V_{h+1}^\tau \psi(s_h, a_h, s')^\top \theta_h \mathrm{d}s'\right] = (\phi_h^\tau)^\top \theta_h, \quad (5.1)$$

where the first equality uses the definition of the operator $\mathbb{P}_h$ defined in (2.2), the second equality follows from the definition of linear mixture MDP in Definition 2.1, and the last equality is obtained by the definition that $\phi_h^\tau = \mathbb{E}_{\pi^\tau}[\phi_{V_{h+1}^\tau}(s_h, a_h)]$. Here $\mathbb{E}_{\pi^\tau}$ is taken respect to $(s_h, a_h)$ and we omit $(s_h, a_h)$ for simplicity. Inspired by (5.1), for any $h \in [H]$, we solve the following weighted ridge regression problem to estimate $\theta_h$

$$\bar{\theta}_h^k \leftarrow \arg\min_\theta \sum_{\tau=1}^{k-1} \frac{\left((\widehat{\phi}_h^\tau)^\top \theta - y_h^\tau\right)^2}{w_\tau^2} + \lambda \|\theta\|_2^2, \quad (5.2)$$

where the weight $w_\tau$ is calculated in previous $k-1$ episodes and we will specify its choice in Line 6 of Algorithm 1. $\lambda$ is a regularization parameter that will be specified in Theorem 5.2. The solution of (5.2) takes the form

$$\bar{\theta}_h^k = (\Lambda_h^k)^{-1}\left(\sum_{\tau=1}^{k-1} \frac{\widehat{\phi}_h^\tau \cdot y_h^\tau}{w_\tau^2}\right), \quad \text{where } \Lambda_h^k = \sum_{\tau=1}^{k-1} \frac{\widehat{\phi}_h^\tau (\widehat{\phi}_h^\tau)^\top}{w_\tau^2} + \lambda I_d. \quad (5.3)$$

We remark that the work (Zhou et al., 2021) in classical RL also adopted the weighted ridge regression to estimate models. The $w_\tau^2$ in Zhou et al. (2021) is the estimated variance, while $w_\tau^2$ here is the estimation uncertainty measured by matrix weighted norm (cf. Line 6 of Algorithm 1). Besides, Wan et al. (2022) applied a similar weighted ridge regression to quantum linear bandits. However, we have an extra challenge in determining $w_\tau$ to incorporate the quantum speed-up: the feature $\phi_h^\tau$ is unknown and needs to be estimated from data. To this end, we propose the "feature estimation" part, which is completely new and essential for RL. See Remark 5.1 for details.

**Optimistic planning.** Given the estimators $\{\bar{\theta}_h^k\}_{h\in[H]}$, we construct the confidence set $\mathcal{C}^k = \{\mathcal{C}_h^k\}$ for $\{\theta_h\}_{h\in[H]}$, where

$$\mathcal{C}_h^k = \left\{ \theta : \|\theta - \bar{\theta}_h^k\|_{\Lambda_h^k} \le \beta_k \right\}, \tag{5.4}$$

and $\beta_k \ge 0$ is the radius of the confidence set specified later. We will prove that $\theta_h \in \mathcal{C}_h^k$ for all $h \in [H]$ with high probability. Based on this confidence set, we can perform the optimistic planning. See Algorithm 5 in Appendix F for details.

**Feature estimation.** Recall that we use estimated features $\{\widehat{\phi}_h^\tau\}_{(\tau,h)\in[k-1]\times[H]}$ to perform the weighted ridge regression in (5.2) since $\{\phi_h^\tau\}_{(\tau,h)\in[k-1]\times[H]}$ are unknown. In the $k$-th phase, we need to estimate features $\{\phi_h^k\}_{h\in[H]}$ by quantum tools. **For the sake of theoretical analysis (cf. Appendix F), the ideal estimators $\{\widehat{\phi}_h^k\}_{h\in[H]}$ should satisfy**

$$\|\widehat{\phi}_h^k - \phi_h^k\|_2 \le \max_{h\in[H]} \|\widehat{\phi}_h^k\|_{(\Lambda_h^k)^{-1}}, \tag{5.5}$$

**for all $h \in [H]$. Even for the classical setting, this problem still seems challenging since the accuracy in the right hand side of (5.5) depends on the estimators $\{\widehat{\phi}_h^k\}_{h\in[H]}$ themselves. Meanwhile, we hope to achieve acceleration in estimating $\{\phi_h^k\}_{h\in[H]}$ with the help of quantum tools, which poses another challenge. To this end, we propose a novel feature estimation process, which leverages a novel combination of the binary search and the quantum mean estimation oracle. See Algorithm 2 and Appendix D.2 for details.**

**Remark 5.1.** Since linear bandits do not involve unknown transition kernels and the feature is known to the learner (Wan et al., 2022), this challenge is unique to linear mixture MDPs. Meanwhile, in classical RL (Ayoub et al., 2020), there is no need to estimate the feature due to martingale analysis, which typically results in only a $\sqrt{T}$ regret, without a quantum computing speedup counterpart. Finally, we would like to emphasize that the technical challenge elaborated in (5.5) has not appeared in previous literature, and the corresponding algorithm design (Algorithm 2) and analysis (e.g., Lemma F.3) are entirely new. More elaborations of our novelties are deferred to Appendix B.2.

**Regression targets estimation.** At the end of $k$-th phase, we use quantum multivariate mean estimation oracle in Lemma 3.2 to calculate regression targets $\{y_h^k\}_{h\in[H]}$. We defer the details to Appendix D.3 and Algorithm 6 in Appendix F.

The theoretical guarantee for Algorithm 1 is given below, and the proof is deferred to Appendix F.

**Theorem 5.2.** Let $\lambda = 1$ in (5.3) and $\beta_k = 1 + 2\sqrt{dk}$ in (5.4). Fix $\delta > 0$. Then with probability at least $1 - \delta$, the regret bound of Algorithm 1 satisfies

$$\text{Regret}(T) = \mathcal{O}\left( d^{5/2} H^{9/2} \log^{3/2} \left(1 + \frac{T^3}{d}\right) \cdot \iota \right),$$

where $\iota = \mathcal{O}(\log(dH\log(1 + T^3/d)/\delta))$.

We have established an $\mathcal{O}(\text{poly}(d, H, \log T))$ regret in Theorem 5.2 as desired. This logarithmic regret breaks the $\Omega(\sqrt{T})$ barrier in classical RL (Zhou et al., 2021).

**Proof Sketch.** Firstly, we show that the phase number $K$ is bounded by $\widetilde{\mathcal{O}}(dH)$ (Lemma F.1). Additionally, we establish that the $k$-th phase contains at most $\widetilde{\mathcal{O}}(\sqrt{d}H^2 / \max_{h\in[H]} \|\widehat{\phi}_h^k\|_{(\Lambda_h^k)^{-1}})$ episodes, as detailed in Lemma F.3 and Lemma F.4. Through a novel regret decomposition analysis (Lemma F.2 and the proof in Appendix F.6), we establish that each episode in the $k$-th phase incurs at most $\widetilde{\mathcal{O}}(H\beta_k \cdot \max_{h\in[H]} \|\widehat{\phi}_h^k\|_{(\Lambda_h^k)^{-1}})$ error. By combining these results, we derive our final regret bound. Notably, our proof diagram, especially the feature estimation analysis, is completely new.

## 6 CONCLUSION

In this paper, we initiate the study of the online exploration problem in quantum RL. We propose a novel algorithm for tabular MDPs and prove that the proposed algorithm enjoys a logarithmic regret. Furthermore, we extend our results to the linear function approximation setting, which is capable of handling problems with large state space. To our best knowledge, we provide the first theoretical understanding of online exploration in quantum RL.

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

# A  ADDITIONAL RELATED WORKS

Exploration is the core problem in online RL. In the literature of classical RL, there is a long line of works designing no-regret algorithms for tabular RL (Jaksch et al., 2010; Azar et al., 2017; Jin et al., 2018; Zanette & Brunskill, 2019; Zhang et al., 2021; Wu et al., 2022; Zhang et al., 2022), RL with linear function approximation (Yang & Wang, 2019; Ayoub et al., 2020; Jin et al., 2020; Cai et al., 2020; Yang & Wang, 2020; Modi et al., 2020; Zhou et al., 2021), and RL with general function approximation (Jiang et al., 2017; Sun et al., 2019; Wang et al., 2020; Du et al., 2021; Jin et al., 2021; Foster et al., 2021; Zhong et al., 2022; Chen et al., 2022). Our work is mostly related to Jaksch et al. (2010) and Ayoub et al. (2020). Specifically, Jaksch et al. (2010) proposed the UCRL2 algorithm for tabular RL and established an $\widetilde{\mathcal{O}}(\mathrm{poly}(S, A, H) \cdot \sqrt{T})$ regret bound. Ayoub et al. (2020) focused on linear mixture MDPs and proposed the UCRL-VTR algorithm with $\widetilde{\mathcal{O}}(\mathrm{poly}(d, H) \cdot \sqrt{T})$ regret bound, where $d$ is the ambient dimension. In comparison, we leverage quantum tools and design novel algorithms with logarithmic regret for both tabular and linear mixture MDPs.

For classical RL, Jaksch et al. (2010); Jin et al. (2018); Zhou et al. (2021) showed that, without additional assumption, the $\sqrt{T}$ regret is inevitable. To this end, some works (Simchowitz & Jamieson, 2019; Yang et al., 2021; He et al., 2021; Xu et al., 2021; Dong & Ma, 2022) proposed algorithms with logarithmic gap-dependent regret. In contrast, we establish logarithmic worst-case regret in quantum RL.

Quantum machine learning has become a popular research topic in recent years (Schuld et al., 2015; Biamonte et al., 2017; Arunachalam & de Wolf, 2017; Dunjko & Briegel, 2018). In particular, the study of quantum RL is rapidly advancing (see e.g., Meyer et al., 2022). In the following, we make comparisons to existing literature in quantum RL related to our work.

- Quantum bandits: For the special model of bandits, Casalé et al. (2020); Wang et al. (2021b) studied the exploitation problem of best-arm identification of multi-armed bandits. As for exploration, Wan et al. (2022) proposed quantum algorithms for multi-armed bandits and linear bandits with logarithmic regret, and Li & Zhang (2022) extended logarithmic regret guarantee to quantum algorithms for stochastic convex bandits. Lumbreras et al. (2022) studied exploration versus exploitation when learning properties of quantum states. We note that these papers focused on the bandit problems and are not as general as the tabular and linear mixture MDPs studied in this paper.

- Quantum policy iteration: Cherrat et al. (2022); Wiedemann et al. (2022) studied how to train MDPs on quantum computers via policy iteration, and Cornelissen (2018); Jerbi et al. (2022) studied quantum algorithms for computing policy gradients. These papers focused on how to train MDPs on quantum computers and did not study the exploration guarantee of quantum algorithms.

- Quantum RL with quantum environments: Dunjko et al. (2015; 2016; 2017); Hamann et al. (2021); Saggio et al. (2021); Hamann & Wölk (2022) considered genuine quantum environments and corresponding quantum RL models. These results exploit RL to solve quantum problems, while we focus on classical RL problems and use quantum computing to make their exploration more efficient.

**Comparison with Concurrent Work (Ganguly et al., 2023).** The concurrent and independent work (Ganguly et al., 2023) also focuses on quantum RL and establishes logarithmic regret for **tabular** MDPs. We provide comparisons between Ganguly et al. (2023) and our work as follows.

- The problem settings are different. Ganguly et al. (2023) assumes an MDP proceeding classically, with an agent providing a certain action $a_h$ at time step $h$, getting reward $r_h$, and stepping to next state $s_{h+1}$ from $s_h$. Here $(s_h, a_h, r_h, s_{h+1})$ are collected as classical information. For each step, the agent is assumed to observe $S$ additional quantum samples returned by $S$ unitaries: $U_s|0\rangle = \sqrt{1 - \mathcal{P}_h(s_{h+1} = s \mid s_h, a_h)}|0\rangle + \sqrt{\mathcal{P}_h(s_{h+1} = s \mid s_h, a_h)}|1\rangle, \forall s \in \mathcal{S}$, which encode the probability of making transition to $s_{h+1}$ by taking action $a_h$ at state $s_h$ in order to estimate the distribution $\mathcal{P}_h(s_{h+1} = \cdot \mid s_h, a_h) \in \mathbb{R}^S$. In contrast, our work considers a quantum-accessible environment. In our work, the agent is allowed to take a quantum policy as (3.4), which is stochastic and is naturally evaluated by unitaries. The transition and reward are also given by quantum unitaries as (3.2) and (3.3). The whole episode of MDP is quantized and it is natural to quantum computers. Due the the different models studied by Ganguly et al. (2023) and us, our regret

bound is worse than their regret bound by a factor of $H$. We want to emphasize that our analysis immediately implies the same regret bound as theirs under their setting.

- More importantly, our work also considers linear mixture MDPs, which are not considered by Ganguly et al. (2023). This setting is significantly more challenging than the tabular setting. To establish the logarithmic regret for linear mixture MDPs, we develop new algorithm designs and theoretical analysis, as detailed in the "Challenges and Technical Overview" part of the introduction section, Section 5, and Appendix B.2.

# B  ADDITIONAL DISCUSSIONS

## B.1  DISCUSSIONS ON REGRET LOWER BOUND AND SAMPLE COMPLEXITY

We first discuss the relationship between regret guarantee and sample complexity. Similar arguments have been presented in Jin et al. (2018).

- **A regret upper bound guarantee implies a sample complexity upper bound.** If an algorithm executes policies $\{\pi_i\}_{i=1}^T$ and incurs a regret $\mathrm{Reg}(T)$ in $T$ episodes, the algorithm can output a policy $\bar{\pi} = \mathrm{Uniform}(\{\pi_i\}_{i=1}^T)$ (uniformly select a policy). The gap between $\bar{\pi}$ and the optimal policy $\pi^*$ is $\mathrm{Reg}(T)/T$. Solving the inequality $\mathrm{Reg}(T)/T \leq \epsilon$ gives the sample complexity of the algorithm. For example, a classical $\sqrt{T}$-regret upper bound implies a $\epsilon^{-2}$ sample complexity; our $\log T$-regret upper bound implies a $\epsilon^{-1}$ sample complexity.

- **A sample complexity lower bound implies a regret lower bound guarantee.** Assuming $g(\epsilon)$ is a known sample complexity lower bound, for any algorithm with regret $\mathrm{Reg}(T)$, it implies a sample complexity $f(\epsilon)$ as discussed above. Solving the inequality $f(\epsilon) \geq g(\epsilon)$ yields a regret lower bound. For example, suppose there is a hard instance where every algorithm necessitates $C/\epsilon$ (resp. $C/\epsilon^2$) sample complexity to obtain the $\epsilon$ optimal policy, where $C$ involves certain problem parameters (e.g., $S, A, H, d$) and sufficiently large constants. Now, assume the existence of an algorithm that achieves regret $o(C)$ (resp. $o(\sqrt{CT})$). This implies an $o(C/\epsilon)$ (resp. $o(C/\epsilon^2)$) sample complexity, which contradicts the sample complexity lower bound. Hence, we can conclude that in this hard instance, every algorithm incurs a regret at least $\Omega(C)$ (resp. $\Omega(\sqrt{CT})$).

**Regret lower bound.**  Since the MDP with a generative model is a simpler setting than the online quantum RL considered by us, the sample complexity lower bound $\Omega(S\sqrt{A}H^{1.5}/\epsilon)$ in the previous work Wang et al. (2021b) immediately implies the sample complexity lower bound for online quantum RL. As discussed above, this $\Omega(S\sqrt{A}H^{1.5}/\epsilon)$ further implies a $\Omega(S\sqrt{A}H^{1.5})$ regret lower bound ($C = S\sqrt{A}H^{1.5}$ therein). Regarding the lower bound for linear mixture MDPs, we can regard the hard instance constructed in [1] as a linear mixture MDP with $d = S^2 A$ (see also our discussions below (2.2)). Hence, by the same proof in Wang et al. (2021b), we can obtain an $\Omega(\sqrt{dH^3}/\epsilon)$ sample complexity lower bound and an $\Omega(\sqrt{dH^3})$ regret lower bound for linear mixture MDPs.

**Further discussions on the gap between upper bounds and lower bounds.**  Ignoring the logarithmic terms (in $T$ or $\epsilon$), our regret/sample complexity can be improved in terms of $(S, A, H)/(d, H)$. We conjecture that all these dependencies can be enhanced. One potential approach is using the variance reduction technique, as in classical online RL (Azar et al., 2017; Zhou et al., 2021). However, we currently lack an understanding of how to apply this in the context of online quantum RL. Moreover, deriving the minimax optimal sample complexity in MDPs with a generative model remains an open question, and addressing the sample complexity problem in this simpler setting seems more feasible and may provide more insights into obtaining sharper bounds in online quantum RL. As an initial exploration of online quantum RL, we leave achieving the tighter or even minimax optimal regret bound as future work.

## B.2  MORE DISCUSSIONS ON OUR NOVELTIES

**Algorithm design novelties:**  One can regard the bandits as the MDPs satisfying (i) the state space only contains a single dummy state $s_{\mathrm{dummy}}$; (ii) $H = 1$; and (iii) the reward only depends on the action chosen by the learner, i.e., $r(s_{\mathrm{dummy}}, a) = r(a)$. In this case, the learner can repeatedly

pull a particular arm $a$ to collect sufficient samples to estimate $r(s_{\text{dummy}}, a)$. However, in online RL, the state will transit according to the transition kernel, preventing repeated arm pulls for the desired estimator. To address this, we introduce a novel adaptive lazy-updating rule, quantifying the uncertainty of the visiting state-action pair and updating the model infrequently through the doubling trick. This algorithm design, absent in previous works on quantum bandits [1] and quantum RL with a generative model [2], connects online exploration in quantum RL with strategic lazy updating mechanisms, inspiring subsequent algorithm design in online quantum RL.

**Technical novelties:** In the **tabular** setting, a significant technical challenge is obtaining a high probability regret bound, as vanilla regret decomposition (e.g., the one used in UCBVI) leads to a martingale term of order $\mathcal{O}(H\sqrt{T})$. Though it is not the dominating term in the classical setting, it would become the dominating term in our setting. We found that another regret decomposition (i.e., the one used in EULER) based on the expected Bellman error cleverly bypasses such martingale terms in the regret, thus achieving the desired regret bound.

In the **linear** setting, the algorithm requires a novel design to leverage the advantage of quantum mean estimation. Classical linear RL analysis heavily relies on martingale concentration, such as the well-known self-normalized concentration bound for vectors. However, in quantum RL, there is no direct counterpart to martingale analysis. Consequently, we redesign the lazy updating frequency and propose an entirely new technique to estimate features for building the confidence set for the model. Notably, previous works on quantum bandits (Wan et al., 2022) do not need to take this step since there is no unknown transition kernel in their setting. Moreover, estimating the features poses a subtle technical challenge, as elaborated in (5.5). Our proposed algorithm (Algorithm 2), which features binary search and quantum mean estimation, successfully addresses this technical challenge. Meanwhile, the quantum samples used in feature estimation approximately equal the number of quantum samples in each phase, eliminating extra regret for this additional feature estimation. This algorithm design (Algorithm 2) and theoretical analysis are entirely new in the literature on (classical and quantum) RL theory.

## C ADDITIONAL EXPLANATIONS ON QUANTUM COMPUTING

In this section, we supplement the details about the basics of quantum computing, and how to simulate classical sampling in quantum-accessible environments.

### C.1 ORACLES

First, we give the explicit definition about the probability oracle and binary oracle mentioned in Lemma 3.1 and Lemma 3.2.

**Definition C.1** (Probability Oracle). Consider a probability distribution $p$ on a finite set $\Omega$. We say $U_p$ is a probability oracle for $p$ if

$$U_p \colon |0\rangle \to \sum_{\omega \in \Omega} \sqrt{p(\omega)} |\omega\rangle |\phi_\omega\rangle$$

where $\{|\omega\rangle\}_{\omega \in \Omega}$ are orthonormal vectors representing $\omega \in \Omega$, and $|\phi_\omega\rangle$ are ancilla quantum states. Commonly we set $\Omega = \{1, \ldots, n\}$ for some integer $n$.

**Definition C.2** (Binary Oracle). Consider a random variable $X \colon \Omega \to \mathbb{R}$ on a finite probability space $(\Omega, p)$. We say $U_X$ is a binary oracle for $X$ if

$$U_X \colon |\omega\rangle |0\rangle \to |\omega\rangle |X(\omega)\rangle \quad \forall \omega \in \Omega$$

In the above definitions $|0\rangle$ can be short for $|0 \ldots 0\rangle$ with more than one qubits.

**Remark C.3.** A quantum oracle is not only applied to $|0\rangle$ or the computational basis. It is a unitary that acts on any quantum state in the corresponding Hilbert space. For example, the binary oracle $U_X$ can be used as below:

$$\sum_{\omega \in \Omega} a_\omega |\omega\rangle |0\rangle \xrightarrow{U_X} \sum_{\omega \in \Omega} a_\omega |\omega\rangle |X(\omega)\rangle$$

where $\sum_{\omega \in \Omega} a_\omega |\omega\rangle$ is a normalized quantum state, i.e., $\sum_{\omega \in \Omega} |a_\omega|^2 = 1$.

## C.2 CLASSICAL SAMPLING VIA QUANTUM ACCESS

We introduce the subroutine Classical Sampling via Quantum Access (CSQA) here (Jerbi et al., 2022; Wiedemann et al., 2022). On the input policy $\pi$ and target step $h$, CSQA returns a sample $s_h \sim d_h^\pi$ using at most 1 episode of quantum interactions (one call to each oracle $\bar{\mathcal{P}}_{h'}$ for $h' < h$). Note that according to Grover & Rudolph (2002) we know the unitary $\Pi_h$ is efficiently computable given a classical policy $\pi = \{\pi_h\}_{h=1}^H$ for any $h \in [H]$.

---

**Algorithm 3** Classical Sampling via Quantum Access (CSQA)

---

1: **Input:** policy $\pi$, target step $h$.
2: Prepare $|\varphi\rangle := |s_1\rangle|0\rangle_{\mathcal{A}}(|0\rangle_{\mathcal{S}}|0\rangle_{\mathcal{A}})^{\otimes(h-2)}|0\rangle_{\mathcal{S}}$.
3: **for** $h' = 1, 2, ..., h-1$ **do**
4:     Call $\Pi_{h'}$ on the $(2h'-1)$-th and $2h'$-th register of $|\varphi\rangle$.
5:     **if** $h' < h-1$ **then**
6:         Call $\bar{\mathcal{P}}_{h'}$ on the $(2h'-1)$-th, $2h'$-th, and $(2h'+1)$-th register of $|\varphi\rangle$.
7:     **end if**
8: **end for**
9: **Output:** Measure the last register of $|\varphi\rangle$ in the standard basis of $\bar{\mathcal{S}}$, and output the result.

---

Here the notation $|0\rangle_{\mathcal{S}}$ and $|0\rangle_{\mathcal{A}}$ are used to distinguish the superposition of states and actions, $\otimes$ denotes the tensor product. The implementation of CSQA is simply a quantum simulation of the classical sampling procedure. We use $|\varphi_{h'}\rangle$ to denote the $(2h'-1)$-th register of $|\varphi\rangle$. Starting with $h' = 1$, CSQA performs the quantum evaluation of $\pi_{h'}$ on $|\varphi_{h'}\rangle$ and get the result $\Pi_{h'}|\varphi_{h'}\rangle|0\rangle_{\mathcal{A}}$ at first. Then CSQA calls $\bar{\mathcal{P}}_{h'}$ on $\Pi_{h'}|\varphi_{h'}\rangle|0\rangle_{\mathcal{A}}|0\rangle_{\mathcal{S}}$ to obtain $|\varphi_{h'+1}\rangle$ as the last register. It is straightforward to verify the correctness of this process: if

$$|\varphi_{h'}\rangle = \sum_s \sqrt{d_{h'}^\pi(s)}|s\rangle,$$

by the definition in (3.2), we know the last register of $\bar{\mathcal{P}}_{h'}\Pi_{h'}|\varphi_{h'}\rangle|0\rangle_{\mathcal{A}}|0\rangle_{\mathcal{S}}$ (i.e., $|\varphi_{h'+1}\rangle$) equals

$$|\varphi_{h'+1}\rangle = \sum_s \sqrt{d_{h'+1}^\pi(s)}|s\rangle,$$

since[4]

$$\Pi_{h'}|\varphi_{h'}\rangle|0\rangle_{\mathcal{A}}|0\rangle_{\mathcal{S}} = \sum_{s,a} \sqrt{d_{h'}^\pi(s,a)}|s,a\rangle|0\rangle_{\mathcal{S}} \xrightarrow{\bar{\mathcal{P}}_{h'}} \sum_{s,a,s'} \sqrt{d_{h'}^\pi(s,a)\mathcal{P}_h(s'\mid s,a)}|s,a,s'\rangle$$

be definition. Therefore, the last register $s'$ is equal to

$$\sum_{s'} \sqrt{\sum_{s,a} d_{h'}^\pi(s,a)\mathcal{P}_h(s'\mid s,a)}|s'\rangle = \sum_{s'} \sqrt{d_{h'+1}^\pi(s')}|s'\rangle = |\varphi_{h'+1}\rangle.$$

Finally, we can measure the last register of $|\varphi\rangle$ and obtain a sample $|s_h\rangle$ with probability $d_h^\pi(s_h)$.

**Remark C.4** (Discussion on Counter Updating). First, we would like to clarify that tracking the number of quantum samples $n_h(s,a)$ does not require the agent to know the next state, whether it is a classical state or a quantum state. Tracking $n_h(s,a)$ only requires knowing the classical state $(s,a)$ at step $h$, which is achieved by CSQA (Algorithm 3). This algorithm returns a classical state $(s,a)$ at step $h$ according to the given roll-out policy. When we use $(s,a)$ to query the transition oracle at step $h$, it is equivalent to apply $\bar{\mathcal{P}}_h$ on the input state $|s,a\rangle|0\rangle$, since $|s,a\rangle$ denotes the quantum representation of $(s,a)$ in the Hilbert space of quantum superpositions. A quantum sample is returned after the query, so we can increase the counter $n_h(s,a)$ by 1 since we gain a new independent quantum sample $\bar{\mathcal{P}}_h|s,a\rangle|0\rangle$.

---

[4]With a little abuse of notations, we use $d_h^\pi(\cdot,\cdot)$ to denote the occupancy measure of $\pi$ at step $h$ over state-action space.

# D  QUANTUM ORACLES USED IN THE ALGORITHMS

For the tabular MDPs, we use the quantum probability oracles for $\mathcal{P}_h(\cdot \mid s, a)$ for each $(s, a, h) \in \mathcal{S} \times \mathcal{A} \times [H]$ to construct estimators $\widehat{\mathcal{P}}_h(\cdot \mid s, a)$. For the linear mixture MDPs, we construct the quantum probability oracles and quantum binary oracles for the feature $\phi_h^k$ and the regression target $\mathbb{E}_{\pi^k}[\mathbb{P}_h V_{h+1}^k(s_h, a_h)]$ to estimate them efficiently. In this section, we describe how to construct these quantum probability oracles in detail. Note that in our case, the quantum oracle encodes a classical distribution; for simplicity, we omit ancilla quantum states in probability oracles in Definition C.1. The simplified probability oracle is also sufficient for estimation (van Apeldoorn, 2021; Cornelissen et al., 2022). See also Gilyén & Li (2020) for the relationships between quantum oracles that encode distributions.

## D.1  MODEL ESTIMATION IN TABULAR MDPS

The only usage of the quantum probability oracles in Quantum UCRL (Algorithm 4) is at Line 11, when the number of visits $n_h(s_h, a_h)$ reaches $2^{l_h(s_h, a_h)}$ for step $h$ and some state-action pair $(s_h, a_h)$. At this time, the dataset $\mathcal{D}_h(s_h, a_h)$ stores all the quantum samples querying $\bar{\mathcal{P}}_h$ on the same input $|s_h, a_h\rangle|0\rangle$. By definition of $\bar{\mathcal{P}}_h$, this is equivalent to the quantum samples obtained by querying the following quantum probability oracle

$$U_{h, s_h, a_h} : |0\rangle \to \sum_{s'} \sqrt{\mathcal{P}_h(s' \mid s_h, a_h)} |s'\rangle. \tag{D.1}$$

Note that we have $n_h(s_h, a_h)$ independent quantum samples obtained by calling $U_{h, s_h, a_h}$ (or its inverse). Therefore, we can turn to the quantum multi-dimensional amplitude estimation subroutine (Lemma 3.1) to obtain an estimate $\widehat{\mathcal{P}}_h(\cdot \mid s_h, a_h)$ such that

$$\left\| \widehat{\mathcal{P}}_h(\cdot \mid s_h, a_h) - \mathcal{P}_h(\cdot \mid s_h, a_h) \right\|_1 \le \frac{c_1 S}{n_h(s_h, a_h)} \log \frac{S}{\delta} \tag{D.2}$$

with probability at least $1 - \delta$.

**Remark D.1** (Discussion on Binary Oracle in Tabular MDPs)**.** We can definitely generalize the results of Lemma 3.1 to Lemma 3.2 with binary oracle in the tabular setting. This is done in the following way. For a fixed $(s, a)$ pair, we define the binary oracle $U_{s,a} : |s, a\rangle|s'\rangle|0\rangle \to |s, a\rangle|\vec{1}[s']\rangle$, where $\vec{1}[s']$ is an $S$-dimensional standard basis vector encoded by $s'$ (i.e., a one-hot vector with non-zero entry at the $s'$ position). Combined with the probability oracle $\bar{\mathcal{P}}_h : |s, a\rangle|0\rangle \to \sum_{s'} \sqrt{\mathcal{P}_h(s' \mid s, a)} |s'\rangle$, we can estimate the $S$-dimensional vector $(\mathcal{P}_h(s_1 \mid s, a), \mathcal{P}_h(s_2 \mid s, a), ..., \mathcal{P}_h(s_S \mid s, a))^\top \in \mathbb{R}^S$ suppose $s'$ is enumerated from the state space $\{s_1, s_2, ..., s_S\}$ using Lemma 3.2. Since we hope the $l_1$ estimation error to be bounded by $\epsilon$, the sample complexity of such estimation is $\mathcal{O}(S \log(S/\delta)/\epsilon)$, the same as Lemma 3.1. Since the target vector $(\mathcal{P}_h(s_1 \mid s, a), \mathcal{P}_h(s_2 \mid s, a), ..., \mathcal{P}_h(s_S \mid s, a))$ is actually a distribution over the state space, we can also encode this vector as the amplitude of a quantum superposition and use Lemma 3.1 to estimate the amplitude. We choose Lemma 3.1 mainly because the quantum multi-dimensional amplitude estimation subroutine is conceptually simpler without requiring an additional binary oracle, so it is more efficient in implementation.

## D.2  FEATURE ESTIMATION IN LINEAR MIXTURE MDPS

Recall that we need to estimate the feature

$$\phi_h^k = \mathbb{E}_{\pi^k} \left[ \phi_{V_{h+1}^k}(s_h, a_h) \right] \tag{D.3}$$

in order to perform the value target regression in the phase $k$ of Quantum UCRL-VTR (Algorithm 1). Here $\phi_{V_{h+1}^k}(\cdot, \cdot)$ is a known feature mapping. Define the binary oracle $U_{V, k, h+1}$ for $\phi_{V_{h+1}^k}(\cdot, \cdot)$ as follows

$$U_{V, k, h+1} : |s, a\rangle|0\rangle \to |s, a\rangle|\phi_{V_{h+1}^k}(s, a)\rangle. \tag{D.4}$$

Note that as $\phi_{V_{h+1}^k}(\cdot,\cdot)$ is entirely known, we can construct this binary oracle by applying a series of $SA$ controlled gates on the two registers, where the $i$-th controlled gate ($i \in [SA]$) applies an addition of $\phi_{V_{h+1}^k}(s,a)$ to the second register if and only if the first register equals to the $i$-th pair of $(s,a)$ among all $SA$ possibilities. Each of these gates can be implemented on $\mathcal{O}(\log SA)$ qubits with gate complexity $\mathcal{O}(\log SA)$ (Barenco et al., 1995).

Next, we construct the quantum probability oracle

$$U_{k,h} : |0\rangle \to \sum_{s,a} \sqrt{d_h^{\pi^k}(s,a)}|s,a\rangle \tag{D.5}$$

by the same way as in the CSQA subroutine (Algorithm 3), where we do not measure $\varphi_h$ at the last line of CSQA and outputs $\Pi_h^k \varphi_h |0\rangle$ instead for $\Pi_h^k$ being the quantum evaluation of $\pi_h^k$.

Now equipped with the quantum probability oracle $U_{k,h}$ and binary oracle $U_{V,k,h+1}$, we can construct the estimation of $\phi_h^k$ by the quantum multivariate mean estimation subroutine (Lemma 3.2) to build estimators $\widehat{\phi}_h^k$ as in Algorithm 2.

### D.3 REGRESSION TARGET ESTIMATION IN LINEAR MIXTURE MDPS

Similar to the feature estimation, we can construct the quantum probability oracle for the regression target

$$\mathbb{E}_{\pi^k}\left[\mathbb{P}_h V_{h+1}^k(s_h, a_h)\right]. \tag{D.6}$$

Note that by definition this term is equal to

$$\mathbb{E}_{\pi^k}\left[V_{h+1}^k(s_{h+1})\right]. \tag{D.7}$$

Therefore, it suffices to construct the binary oracle for $V_{h+1}^k$

$$U'_{V,k,h+1} : |s\rangle|0\rangle \to |s\rangle|V_{h+1}^k(s)\rangle, \tag{D.8}$$

and construct the quantum probability oracle

$$U'_{k,h} : |0\rangle \to \sum_s \sqrt{d_{h+1}^{\pi^k}(s)}|s\rangle. \tag{D.9}$$

These two oracles can be constructed by the same way discussed in the previous section. As a result, we can leverage the quantum mean estimation subroutine (Lemma 3.2) to estimate the regression target as desired in Algorithm 6.

# E PROOFS FOR TABULAR MDPS

## E.1 MISSING ALGORITHM

---
**Algorithm 4** Quantum UCRL
---
1: **Input:** failure probability $\delta$.
2: Initialize for each $\forall (s, a, h) \in \mathcal{S} \times \mathcal{A} \times [H]$:
   - The counter and tag $n_h(s, a) := 0, l_h(s, a) := 0$.
   - The empirical model $\widehat{\mathcal{P}}_h(\cdot \mid s, a) := \text{Unif}(\mathcal{S})$.
   - The set of quantum samples $\mathcal{D}_h(s, a) := \emptyset$.
3: Initialize the exploration policy $\pi^1$ as a uniform policy.
4: **for** phase $k = 1, 2, ..., \lceil T/H \rceil$ **do**
5:      **for** step $h = 1, 2, ..., H$ **do**
6:          Call $s_h^k := \text{CSQA}(\pi^k, h)$ (Algorithm 3), define $a_h^k := \pi_h^k(s_h^k)$.
7:          Define $|\bar{\varphi}_{k,h+1}\rangle := \bar{\mathcal{P}}_h |s_h^k, a_h^k\rangle |0\rangle$ and $|\bar{\varphi}_{k,h+1}^{-1}\rangle := \bar{\mathcal{P}}_h^{-1} |s_h^k, a_h^k\rangle |0\rangle$.
8:          Add $\{|\bar{\varphi}_{k,h+1}\rangle, |\bar{\varphi}_{k,h+1}^{-1}\rangle\}$ to $\mathcal{D}_h(s_h^k, a_h^k)$.
9:          Add the counter $n_h(s_h^k, a_h^k)$ by 1.
10:          **if** $n_h(s_h^k, a_h^k) = 2^{l_h(s_h^k, a_h^k)}$ **then**
11:              Update $\widehat{\mathcal{P}}_h(\cdot \mid s_h^k, a_h^k)$ by calling quantum subroutine (Lemma 3.1) with $\mathcal{D}_h(s_h^k, a_h^k)$.
12:              Add the tag $l_h(s_h^k, a_h^k)$ by 1.
13:              Set $\mathcal{D}_h(s_h^k, a_h^k) = \emptyset$.
14:          **end if**
15:      **end for**
16:      Set $V_{H+1}(s) = 0, \forall s \in \mathcal{S}$.
17:      **for** $h = H, H-1, ..., 1$ **do**
18:          Set bonus term $b_h(s, a) := \widetilde{\mathcal{O}}(HS/n_h(s, a))$.
19:          Update $Q, V$ by (4.2) and (4.3).
20:          Set $\pi_h^{k+1}(s) := \arg\max_a Q_h(s, a)$.
21:      **end for**
22: **end for**
---

## E.2 PROOF

The total number of phases is obviously $K := \lceil T/H \rceil$. Define the surrogate regret as

$$\overline{\text{Regret}}(K) := \sum_{k=1}^{K} V_1^*(s_1) - V_1^{\pi^k}(s_1).$$

Then the cumulative regret is bounded by

$$\text{Regret}(T) \le H \cdot \overline{\text{Regret}}(K)$$

because the policy $\pi^k$ is fixed in phase $k$ and the length of phase $k$ is at most $H$.

To bound the surrogate regret, we first rule out several failure events. Define the failure event $\mathcal{E}$ as

$$\mathcal{E} := \bigcup_{s,a,h} \mathcal{E}_{s,a,h}^1 \bigcup \mathcal{E}^2$$

$$\mathcal{E}_{s,a,h}^1 := \left\{ \exists k \in [K], \left\| \widehat{\mathcal{P}}_h^k(\cdot \mid s, a) - \mathcal{P}_h(\cdot \mid s, a) \right\|_1 > \min\left( \frac{2c_1 SL}{n_h^k(s,a)}, 2 \right) \right\}, \forall (s, a, h) \in \mathcal{S} \times \mathcal{A} \times [H]$$

$$\mathcal{E}^2 := \left\{ \exists (s, a, k, h) \in \mathcal{S} \times \mathcal{A} \times [K] \times [H], n_h^k(s,a) < \frac{1}{2} \sum_{j<k} w_h^j(s,a) - \log\frac{2SAH}{\delta} \right\},$$

where $n_h^k$ denotes the counter $n_h$ at the beginning of phase $k$, $\widehat{\mathcal{P}}_h^k$ denotes the estimator $\widehat{\mathcal{P}}_h$ at the beginning of phase $k$. $L := \log(S^2 AH \log(T)/\delta)$ is a logarithmic term. $w_h^k(s,a) := \Pr_{\pi^k}(s_h = s, a_h = a)$ is the visiting probability of $(s, a)$ at step $h$ by $\pi^k$.

**Lemma E.1** (Failure Event). It holds that $\Pr(\mathcal{E}) \leq \delta$.

*Proof.* Fix any $(s, a, h)$, the estimators $\widehat{\mathcal{P}}_h^k(\cdot \mid s, a)$ switches for at most $\log T$ times during $K$ episodes, because it switches only when $n_h(s, a)$ doubles. According to Appendix D.1, we know for any fixed $k \in [K]$, it holds that

$$\left\| \widehat{\mathcal{P}}_h^k(\cdot \mid s, a) - \mathcal{P}_h(\cdot \mid s, a) \right\|_1 \leq \min\left( \frac{2c_1 SL}{n_h^k(s, a)}, 2 \right)$$

with probability $1 - \delta/2SAH \log(T)$. By union bound we have $\Pr(\mathcal{E}_{s,a,h}^1) \leq \delta/2SAH$. Using another union bound gives $\Pr(\cup_{s,a,h} \mathcal{E}_{s,a,h}^1) \leq \delta/2$.

On the other hand, we know $\Pr(\mathcal{E}^2) \leq \delta/2$ according to Section D.4 of Zanette & Brunskill (2019) or Section B.1 of Dann et al. (2019). As a result, we know $\Pr(\mathcal{E}) \leq \delta$. $\square$

Motivated by the failure event, we formally define the bonus function $b_h(s, a)$ for $(s, a) \in \mathcal{S} \times \mathcal{A}$ as

$$b_h(s, a) := \min\left( \frac{2c_1 HSL}{n_h(s, a)}, 2H \right). \tag{E.1}$$

Next we show the optimism of the value functions.

**Lemma E.2** (Optimism). Denote the optimistic functions $Q$ and $V$ at the beginning of phase $k$ by $Q^k, V^k$. Outside the failure event $\mathcal{E}$ it holds that for all $k \in [K]$

$$V_1^k(s_1) \geq V_1^*(s_1).$$

*Proof.* Let the bonus function $b_h$ at the beginning of phase $k$ be $b_h^k$. We prove this lemma by induction on $h = H, H - 1, ..., 1$.

Note that $Q_H^k(s, a) = Q_H^*(s, a) = \mathcal{R}_h(s, a)$ by definition. Thus, $V_H^k(s) \geq V_H^*(s)$ for all $s \in \mathcal{S}$.

Suppose $V_{h'}^k(s) \geq V_{h'}^*(s)$ for all $h' > h$ and $s \in \mathcal{S}$. By definition we have

$$
\begin{aligned}
Q_h^*(s, a) - Q_h^k(s, a) &= \mathbb{P}_h V_{h+1}^*(s, a) - \widehat{\mathbb{P}}_h^k V_{h+1}(s, a) + b_h^k(s, a) \\
&\leq \mathbb{P}_h V_{h+1}^*(s, a) - \widehat{\mathbb{P}}_h^k V_{h+1}^*(s, a) + b_h^k(s, a) \\
&\leq \left\| \mathbb{P}_h - \widehat{\mathbb{P}}_h^k \right\|_1 \cdot \left\| V_{h+1}^* \right\|_\infty + b_h^k(s, a) \\
&\leq 0.
\end{aligned}
$$

The first inequality is due to the induction hypothesis. The last inequality is because $\mathcal{E}$ does not hold and $\|V_{h+1}^*\|_\infty \leq H$. $\square$

Next, we define the "good set" $G_{k,h}$ (Zanette & Brunskill, 2019) of state-action pairs in phase $k$ and step $h$ to be state-action pairs that have been visited sufficiently often.

**Definition E.3.** The good set $G_{k,h}$ for phase $k$ and step $h$ is defined as

$$G_{k,h} := \left\{ (s, a) \in \mathcal{S} \times \mathcal{A} : \frac{1}{4} \sum_{j < k} w_h^j(s, a) \geq \log \frac{2SAH}{\delta} + 1 \right\}.$$

We have the following lemma for $G_{k,h}$.

**Lemma E.4.** Outside the failure event $\mathcal{E}$, for any $(k, h) \in [K] \times [H]$ and $(s, a) \in G_{k,h}$, it holds that

$$n_h^k(s, a) \geq \frac{1}{4} \sum_{j \leq k} w_k^j(s, a).$$

*Proof.* Outside $\mathcal{E}$ we have

$$
\begin{aligned}
n_h^k(s,a) &\geq \frac{1}{2}\sum_{j<k} w_h^j(s,a) - \log\frac{2SAH}{\delta} \\
&= \frac{1}{4}\sum_{j<k} w_h^j(s,a) + \frac{1}{4}\sum_{j<k} w_h^j(s,a) - \log\frac{2SAH}{\delta} \\
&\geq \frac{1}{4}\sum_{j<k} w_h^j(s,a) + 1 \\
&\geq \frac{1}{4}\sum_{j\leq k} w_h^j(s,a).
\end{aligned}
$$

The second inequality is due to the definition of $G_{k,h}$. $\qquad\square$

The state-action pairs outside $G_{k,h}$ contributes little to the regret.

**Lemma E.5.** Define $\bar{L} := \log(2SAH/\delta)$. There exists constant $c_2$ such that

$$
\sum_{k=1}^{K}\sum_{h=1}^{H}\sum_{(s,a)\notin G_{k,h}} w_h^k(s,a) \leq c_2 SAH\bar{L}.
$$

*Proof.* Observe that

$$
\begin{aligned}
\sum_{k=1}^{K}\sum_{h=1}^{H}\sum_{(s,a)\notin G_{k,h}} w_h^k(s,a) &= \sum_{s,a}\sum_{h=1}^{H}\sum_{k=1}^{K} w_h^k(s,a)\mathbf{1}\{(s,a)\notin G_{k,h}\} \\
&\leq \sum_{s,a}\sum_{h=1}^{H}\left(\log\frac{2SAH}{\delta}+1\right) \\
&\leq c_2 SAH\bar{L}.
\end{aligned}
$$

$\qquad\square$

**Lemma E.6** (Rephrased from Lemma 13 of Zanette & Brunskill (2019))**.** For any $h\in[H]$,

$$
\sum_{k=1}^{K}\sum_{(s,a)\in G_{k,h}} \frac{w_h^k(s,a)}{\max(1,n_h^k(s,a))} = \mathcal{O}\left(SA\log T\right).
$$

*Proof of Theorem 4.1.* Outside the failure event $\mathcal{E}$, the surrogate regret can be decomposed as

$$
\begin{aligned}
\overline{\mathrm{Regret}}(K) &= \sum_{k=1}^{K} V_1^*(s_1) - V_1^{\pi^k}(s_1) \leq \sum_{k=1}^{K} V_1^k(s_1) - V_1^{\pi^k}(s_1) \\
&= \sum_{k=1}^{K}\sum_{h=1}^{H}\sum_{s,a} w_h^k(s,a)\cdot\left(\left(\widehat{\mathbb{P}}_h - \mathbb{P}_h\right)V_{h+1}^k(s,a) + b_h^k(s,a)\right) \\
&\leq \sum_{k=1}^{K}\sum_{h=1}^{H}\sum_{s,a} w_h^k(s,a)b_h^k(s,a).
\end{aligned}
$$

The last inequality is by the definition of $\mathcal{E}$ and $b_h^k$.

Using the good set $G_{k,h}$, we further bound the surrogate regret as

$$
\overline{\mathrm{Regret}}(K) \lesssim \sum_{k=1}^{K} \sum_{h=1}^{H} \sum_{s,a} w_h^k(s,a) b_h^k(s,a)
$$

$$
= \sum_{k=1}^{K} \sum_{h=1}^{H} \sum_{(s,a) \in G_{k,h}} w_h^k(s,a) b_h^k(s,a) + \sum_{k=1}^{K} \sum_{h=1}^{H} \sum_{(s,a) \notin G_{k,h}} w_h^k(s,a) b_h^k(s,a)
$$

$$
\leq \sum_{k=1}^{K} \sum_{h=1}^{H} \sum_{(s,a) \in G_{k,h}} w_h^k(s,a) b_h^k(s,a) + 2c_2 H^2 S A \bar{L}.
$$

The last inequality is by Lemma E.5.

The remaining term can be bounded by

$$
\sum_{k=1}^{K} \sum_{h=1}^{H} \sum_{(s,a) \in G_{k,h}} w_h^k(s,a) b_h^k(s,a) \leq \sum_{k=1}^{K} \sum_{h=1}^{H} \sum_{(s,a) \in G_{k,h}} \frac{2c_1 HSL \cdot w_h^k(s,a)}{\max(1, n_h^k(s,a))} + 2H^2 SA
$$

$$
= 2c_1 HSL \sum_{h=1}^{H} \sum_{(s,a) \in G_{k,h}} \sum_{k=1}^{K} \frac{w_h^k(s,a)}{\max(1, n_h^k(s,a))} \mathbf{1}\{(s,a) \in G_{k,h}\} + 2H^2 SA
$$

$$
\lesssim HSL \sum_{h=1}^{H} SA \log(T) + H^2 SA
$$

$$
= H^2 S^2 AL \log(T) + H^2 SA.
$$

The second inequality is by Lemma E.6.

Now we come to our final regret bound

$$
\mathrm{Regret}(T) \leq H \cdot \overline{\mathrm{Regret}}(K) = \mathcal{O}\left( H^3 S^2 AL \log(T) + H^2 SA \bar{L} + H^3 SA \right).
$$

$\square$

## F   Proofs for Linear Mixture MDPs

### F.1   Missing Algorithms

---
**Algorithm 5** Optimistic Planning
---
1: **Input:** Confidence set $\mathcal{C} = \{\mathcal{C}_h\}_{h \in [H]}$.
2: Calculate $\{\widehat{\theta}_h\}_{h \in [H]}$ by solving the following problem:

$$
\underset{\{\theta_h'\}_{h \in [H]}}{\arg\max} V_1'(s_1)
$$

$$
\text{s.t.} \quad \begin{cases} Q_h'(\cdot, \cdot) = R_h(\cdot, \cdot) + \phi_{V_{h+1}'}(\cdot, \cdot)^\top \theta_h', \ \forall h \in [H], \\ V_h'(\cdot) = \max_{a \in \mathcal{A}} Q_h'(\cdot, a), \ \forall h \in [H], \\ V_{H+1}'(\cdot) = 0, \\ \theta_h' \in \mathcal{C}, \ \forall h \in [H]. \end{cases}
$$

3: $V_{H+1}(\cdot) = 0$.
4: **for** $h = H, \cdots, 1$ **do**
5:    $Q_h(\cdot, \cdot) = R_h(\cdot, \cdot) + \phi_{V_{h+1}}(\cdot, \cdot)^\top \widehat{\theta}_h$.
6:    $V_h(\cdot) = \max_{a \in \mathcal{A}} Q_h(\cdot, a)$.
7:    $\pi_h(\cdot \mid \cdot) = \arg\max_\pi \langle Q_h(\cdot, \cdot), \pi_h(\cdot \mid \cdot) \rangle_{\mathcal{A}}$.
8: **end for**
9: **Output:** $(\{Q_h\}_{h \in [H]}, \{V_h\}_{h \in [H]}, \{\pi_h\}_{h \in [H]})$.

---

---

**Algorithm 6** Estimate Regression Target

---

1: **Input:** Policy $\pi$, value functions $\{V_h\}_{h \in [H]}$, and parameter $w$.
2: **for** $h = 1, \cdots, H$ **do**
3:     Calculate $y_h$, the estimator of $\mathbb{E}_\pi[\mathbb{P}_h V_{h+1}(s_h, a_h)]$, by quantum mean estimation subroutine (Lemma 3.2) up to error $w$.
4: **end for**

---

### F.2    Total Number of Phases

**Lemma F.1** (Phase Number). The number of phases $K$ satisfies

$$K \leq \mathcal{O}\Big(dH \cdot \log\big(1 + \frac{T^3}{d}\big)\Big).$$

*Proof of Lemma F.1.* For any $k \in [K]$, there exists an $h_k \in [H]$ such that

$$\|\widehat{\phi}_{h_k}^k\|_{(\Lambda_{h_k}^k)^{-1}} = \max_h \|\widehat{\phi}_h^k\|_{(\Lambda_h^k)^{-1}}. \tag{F.1}$$

By (5.3), we have

$$\Lambda_{h_k}^{k+1} = \lambda I_d + \sum_{\tau=1}^{k} \frac{\widehat{\phi}_{h_k}^\tau (\widehat{\phi}_{h_k}^\tau)^\top}{w_\tau^2} = \Lambda_{h_k}^k + \frac{\widehat{\phi}_{h_k}^k (\widehat{\phi}_{h_k}^k)^\top}{w_k^2}, \tag{F.2}$$

which further implies that

$$
\begin{aligned}
\det(\Lambda_{h_k}^{k+1}) &= \det\Big(\Lambda_{h_k}^k + \frac{\widehat{\phi}_{h_k}^k (\widehat{\phi}_{h_k}^k)^\top}{w_k^2}\Big) \\
&= \det\Big((\Lambda_{h_k}^k)^{1/2}\Big(I_d + \frac{(\Lambda_{h_k}^k)^{-1/2}\widehat{\phi}_{h_k}^k (\widehat{\phi}_{h_k}^k)^\top (\Lambda_{h_k}^k)^{-1/2}}{w_k^2}\Big)(\Lambda_{h_k}^k)^{1/2}\Big) \\
&= \det(\Lambda_{h_k}^k) \cdot \det\Big(I_d + \frac{(\Lambda_{h_k}^k)^{-1/2}\widehat{\phi}_{h_k}^k (\widehat{\phi}_{h_k}^k)^\top (\Lambda_{h_k}^k)^{-1/2}}{w_k^2}\Big).
\end{aligned} \tag{F.3}
$$

By the fact that $\det(I_d + vv^\top) = 1 + \|v\|_2^2$ for any $v \in \mathbb{R}^d$, (F.3) yields

$$\det(\Lambda_{h_k}^{k+1}) = \det(\Lambda_{h_k}^k) \cdot \Big(1 + \frac{\big\|(\Lambda_{h_k}^k)^{-1/2}\widehat{\phi}_{h_k}^k\big\|_2^2}{w_k^2}\Big) = \det(\Lambda_{h_k}^k) \cdot \Big(1 + \frac{\big\|\widehat{\phi}_{h_k}^k\big\|_{(\Lambda_{h_k}^k)^{-1}}^2}{w_k^2}\Big) \tag{F.4}$$

Recall that $w_k = \max_h \|\widehat{\phi}_h^k\|_{(\Lambda_h^k)^{-1}}$. Together with (F.1) and (F.4), we obtain

$$\det(\Lambda_{h_k}^{k+1}) = 2\det(\Lambda_{h_k}^k). \tag{F.5}$$

Meanwhile, by the same argument of (F.2), we know $\Lambda_h^k \preceq \Lambda_h^{k+1}$, which implies that

$$\det(\Lambda_h^{k+1}) \geq \det(\Lambda_h^k), \quad \forall h \in [H]. \tag{F.6}$$

Combining (F.5) and (F.6), we have for any $k \in [K]$

$$\prod_{h=1}^{H} \det(\Lambda_h^{k+1}) \geq 2 \prod_{h=1}^{H} \det(\Lambda_h^k). \tag{F.7}$$

Applying (F.7) to all $k \in [K]$, we have

$$\prod_{h=1}^{H} \det(\Lambda_h^{K+1}) \geq 2^K \prod_{h=1}^{H} \det(\Lambda_h^1) = 2^K \lambda^{dH}, \tag{F.8}$$

where the last equality uses the fact that $\Lambda_h^1 = \lambda \cdot I_d$ for all $h \in [H]$. On the other hand, we have

$$\mathrm{tr}(\Lambda_h^{K+1}) = \mathrm{tr}(\lambda \cdot I_d) + \sum_{\tau=1}^{K} \mathrm{tr}\Big(\frac{\widehat{\phi}_h^\tau (\widehat{\phi}_h^\tau)^\top}{w_\tau^2}\Big) = \lambda d + \sum_{\tau=1}^{K} \frac{\|\widehat{\phi}_h^\tau\|_2^2}{w_\tau^2} \leq \lambda d + \sum_{\tau=1}^{K} \frac{H^2}{w_\tau^2}, \tag{F.9}$$

where the last inequality uses the fact that $\|\widehat{\phi}_h^\tau\|_2 \leq H$ for all $(\tau, h) \in [K] \times [H]$. Since $\Lambda_h^{K+1}$ is positive definite for any $h \in [H]$, we further have

$$\det(\Lambda_h^{K+1}) \leq \Big(\frac{\mathrm{tr}(\Lambda_h^{K+1})}{d}\Big)^d \leq \Big(\lambda + \sum_{\tau=1}^{K} \frac{H^2}{d \cdot w_\tau^2}\Big)^d, \quad \forall h \in [H], \tag{F.10}$$

where the last inequality follows from (F.9). Meanwhile, by the facts that (i) Algorithm 6 uses $(C^\dagger H^2 \cdot \iota / w_\tau)$ trajectories in the $\tau$-th phase (cf. Lemma 3.2), where $C^\dagger$ is an absolute constant; and (ii) the total number of trajectories is $T$, we have $H/w_\tau \leq \mathcal{O}(T)$ for all $\tau \in [K]$. Together with (F.8), (F.10), $\lambda = 1$, and $K \leq T$, we obtain that

$$K \leq \mathcal{O}\Big(dH \cdot \log\big(1 + \frac{T^3}{d}\big)\Big),$$

which finishes the proof of Lemma F.1. $\qquad\square$

### F.3 OPTIMISM

**Lemma F.2** (Optimism). We define the event $\mathcal{E}^\dagger$ as

$$\mathcal{E}^\dagger = \{\theta_h \in \mathcal{C}_h^k, \forall (k, h) \in [K] \times [H]\},$$

where $\mathcal{C}_h^k = \{\theta \in \mathbb{R}^d : \|\theta - \bar{\theta}_h^k\|_{\Lambda_h^k} \leq \beta_k\}$ and $\beta_k = 1 + 2\sqrt{dk}$. Then we have

$$\mathbb{P}(\mathcal{E}^\dagger) \geq 1 - \delta.$$

*Proof of Lemma F.2.* Fix some $(k, h) \in [K] \times [H]$. By the definition of $\bar{\theta}_h$ in (5.3), we have

$$\bar{\theta}_h^k - \theta_h = (\Lambda_h^k)^{-1} \Big(\sum_{\tau=1}^{k-1} \frac{\widehat{\phi}_h^\tau \cdot y_h^\tau}{w_\tau^2}\Big) - \theta_h$$

$$= (\Lambda_h^k)^{-1} \Big(\sum_{\tau=1}^{k-1} \frac{\widehat{\phi}_h^\tau \cdot (y_h^\tau - (\widehat{\phi}_h^\tau)^\top \theta_h)}{w_\tau^2}\Big) - \lambda(\Lambda_h^k)^{-1}\theta_h.$$

For any $x \in \mathbb{R}^d$, by the triangle inequality we have

$$|x^\top(\bar{\theta}_h^k - \theta_h)| \leq \Big|x^\top(\Lambda_h^k)^{-1}\Big(\sum_{\tau=1}^{k-1} \frac{\widehat{\phi}_h^\tau \cdot (y_h^\tau - (\widehat{\phi}_h^\tau)^\top \theta_h)}{w_\tau^2}\Big)\Big| + \lambda \big|x^\top(\Lambda_h^k)^{-1}\theta_h\big|$$

$$\leq \|x\|_{(\Lambda_h^k)^{-1}} \cdot \Big\|\sum_{\tau=1}^{k-1} \frac{\widehat{\phi}_h^\tau \cdot (y_h^\tau - (\widehat{\phi}_h^\tau)^\top \theta_h)}{w_\tau^2}\Big\|_{(\Lambda_h^k)^{-1}} + \lambda\|x\|_{(\Lambda_h^k)^{-1}} \cdot \|\theta_h\|_{(\Lambda_h^k)^{-1}}, \tag{F.11}$$

where the last inequality is obtained by Cauchy-Schwarz inequality. Let $x = \Lambda_h^k(\bar{\theta}_h^k - \theta_h)$, and (F.11) gives that

$$\|\bar{\theta}_h^k - \theta_h\|_{\Lambda_h^k} \leq \underbrace{\Big\|\sum_{\tau=1}^{k-1} \frac{\widehat{\phi}_h^\tau \cdot (y_h^\tau - (\widehat{\phi}_h^\tau)^\top \theta_h)}{w_\tau^2}\Big\|_{(\Lambda_h^k)^{-1}}}_{(i)} + \underbrace{\lambda \cdot \|\theta_h\|_{(\Lambda_h^k)^{-1}}}_{(ii)}. \tag{F.12}$$

**Term (i):** We define

$$\mathbf{M} = \Big(\frac{\widehat{\phi}_h^1}{w_1}, \cdots, \frac{\widehat{\phi}_h^{k-1}}{w_{k-1}}\Big) \in \mathbb{R}^{d \times (k-1)}, \quad \mathbf{N} = \Big(\frac{y_h^1 - (\widehat{\phi}_h^1)^\top \theta_h}{w_1}, \cdots, \frac{y_h^{k-1} - (\widehat{\phi}_h^{k-1})^\top \theta_h}{w_{k-1}}\Big)^\top \in \mathbb{R}^{(k-1) \times 1}. \tag{F.13}$$

With the notations in (F.13), we have

$$(i) = \|\mathbf{MN}\|_{(\Lambda_h^k)^{-1}} = \sqrt{\mathbf{N}^\top \mathbf{M}^\top (\Lambda_h^k)^{-1} \mathbf{MN}} \leq \sqrt{\|\mathbf{N}\|_2 \cdot \|\mathbf{M}^\top (\Lambda_h^k)^{-1} \mathbf{M}\|_2 \cdot \|\mathbf{N}\|_2}, \tag{F.14}$$

where the last inequality follows from Cauchy-Schwarz inequality. Note that

$$
\begin{aligned}
|y_h^\tau - (\widehat{\phi}_h^\tau)^\top \theta_h| &\leq |y_h^\tau - (\phi_h^\tau)^\top \theta_h| + |(\phi_h^\tau)^\top \theta_h - (\widehat{\phi}_h^\tau)^\top \theta_h| \\
&\leq |y_h^\tau - (\phi_h^\tau)^\top \theta_h| + \|\phi_h^\tau - \widehat{\phi}_h^\tau\|_2 \cdot \|\theta_h\|_2 \\
&\leq w_\tau + w_\tau = 2w_\tau,
\end{aligned}
\tag{F.15}
$$

where the first inequality follows from the triangle inequality, the second inequality uses the Cauchy-Schwarz inequality, and the third inequality uses Lemma F.4 and Lemma F.3. Combining (F.15) the definition of $\mathbf{N}$ in (F.13), we have $\|\mathbf{N}\|_\infty \leq 2$, which further implies that

$$
\|\mathbf{N}\|_2 \leq \sqrt{(k-1)} \cdot \|\mathbf{N}\|_\infty \leq 2\sqrt{k}.
\tag{F.16}
$$

Meanwhile, we have

$$
\|\mathbf{M}^\top (\Lambda_h^k)^{-1} \mathbf{M}\|_2 \leq \operatorname{tr}\left(\mathbf{M}^\top (\Lambda_h^k)^{-1} \mathbf{M}\right) = \operatorname{tr}\left((\Lambda_h^k)^{-1} \mathbf{M}\mathbf{M}^\top\right).
\tag{F.17}
$$

By the definition of $\mathbf{M}$ in (F.13), we have

$$
\mathbf{M}\mathbf{M}^\top = \sum_{\tau=1}^{k-1} \frac{\widehat{\phi}_h^\tau (\widehat{\phi}_h^\tau)^\top}{w_\tau^2} = \Lambda_h^k - \lambda \cdot I_d,
\tag{F.18}
$$

where the last equality follows from the definition of $\Lambda_h^k$ in (5.3). Combining (F.17) and (F.18), we have

$$
\|\mathbf{M}^\top (\Lambda_h^k)^{-1} \mathbf{M}\|_2 = \operatorname{tr}\left(I_d - \lambda(\Lambda_h^k)^{-1}\right) \leq \operatorname{tr}(I_d) = d.
\tag{F.19}
$$

Plugging (F.16) and (F.19) into (F.14), we obtain

$$
(\mathrm{i}) \leq 2\sqrt{dk}.
\tag{F.20}
$$

**Term (ii):** By the fact that $\lambda \cdot I_d \preceq \Lambda_h^k$, we have

$$
(\mathrm{ii}) \leq \lambda \cdot \|\theta_h\|_2 / \sqrt{\lambda} \leq \sqrt{\lambda},
\tag{F.21}
$$

where the last inequality uses the fact that $\|\theta\|_2 \leq 1$.

Plugging (F.20) and (F.21) into (F.12), we obtain

$$
\|\bar{\theta}_h^k - \theta_h\|_{\Lambda_h^k} \leq \sqrt{\lambda} + 2\sqrt{dk} = \beta_k,
$$

which concludes the proof of Lemma F.2. $\qquad\square$

### F.4 FEATURE ESTIMATION

**Lemma F.3** (Estimate Feature). Let $\lambda = 1$. It holds with probability at least $1 - \delta/2$ that

$$
\|\widehat{\phi}_h^k - \phi_h^k\|_2 \leq \max_{h \in [H]} \|\widehat{\phi}_h^k\|_{(\Lambda_h^k)^{-1}} = w_k
$$

for all $(k, h) \in [K] \times [H]$. Meanwhile, for each $k \in [K]$, we need at most

$$
\mathcal{O}\left(\frac{\sqrt{d}H^2\iota}{\max_{h \in [H]} \|\phi_h^k\|_{(\Lambda_h^k)^{-1}}}\right)
$$

episodes to calculate $\{\widehat{\phi}_h^k\}_{h \in [H]}$. Here $\iota = \mathcal{O}\left(\log\left(\frac{dH \log(1 + \frac{T^3}{d})}{\delta}\right)\right)$.

*Proof of Lemma F.3.* Fix $h \in [H]$. In Algorithm 2, we construct a series of estimators $\{\widehat{\phi}_{h,n}^k\}_{n=1}^{n_0+1}$ such that

$$
\|\widehat{\phi}_{h,n}^k - \phi_h^k\|_2 \leq 2^{-n}, \quad \forall n \in [n_0 + 1]
\tag{F.22}
$$

until the condition

$$
\|\widehat{\phi}_{h,n_0}^k\|_{(\Lambda_h^k)^{-1}} \geq \frac{1}{2^{n_0-1}},
\tag{F.23}
$$

is satisfied for some fixed $h_0 \in [H]$. Then for any $h \in [H]$ we have

$$\|\widehat{\phi}_{h,n_0+1}^k - \phi_h^k\|_2 \leq 2^{-(n_0+1)}$$
$$\leq \|\widehat{\phi}_{h_0,n_0}^k\|_{(\Lambda_{h_0}^k)^{-1}} - 2^{-n_0} - 2^{-(n_0+1)}, \tag{F.24}$$

where the first inequality uses (F.22) and the second inequality follows from (F.23). By the triangle inequality, we have

$$\|\widehat{\phi}_{h_0,n_0}^k\|_{(\Lambda_{h_0}^k)^{-1}} \leq \|\phi_{h_0}^k\|_{(\Lambda_{h_0}^k)^{-1}} + \|\widehat{\phi}_{h_0,n_0}^k - \phi_{h_0}^k\|_{(\Lambda_{h_0}^k)^{-1}}$$
$$\leq \|\phi_{h_0}^k\|_{(\Lambda_{h_0}^k)^{-1}} + \|\widehat{\phi}_{h_0,n_0}^k - \phi_{h_0}^k\|_2$$
$$\leq \|\phi_{h_0}^k\|_{(\Lambda_{h_0}^k)^{-1}} + 2^{-n_0}, \tag{F.25}$$

where the second inequality uses the fact that $I_d \preceq \Lambda_{h_0}^k$, and the last inequality follows from (F.22). Plugging (F.25) into (F.24), we have for all $h \in [H]$ that

$$\|\widehat{\phi}_{h,n_0+1}^k - \phi_h^k\|_2 \leq \|\phi_{h_0}^k\|_{(\Lambda_{h_0}^k)^{-1}} - 2^{-(n_0+1)}$$
$$\leq \|\widehat{\phi}_{h_0,n_0+1}^k\|_{(\Lambda_{h_0}^k)^{-1}}$$
$$\leq \max_{h \in [H]} \|\widehat{\phi}_{h,n_0+1}^k\|_{(\Lambda_h^k)^{-1}} \tag{F.26}$$

where the second inequality follows from the same argument of (F.25). This finishes the first part of proof.

Meanwhile, following the similar argument of (F.25), we obtain that

$$\|\widehat{\phi}_{h,n_0}^k\|_{(\Lambda_h^k)^{-1}} \geq \|\phi_h^k\|_{(\Lambda_h^k)^{-1}} - 2^{-n_0} \geq 2^{-(n_0-1)}, \tag{F.27}$$

where the last inequality holds when

$$2^{n_0} \geq \frac{3}{\|\phi_h^k\|_{(\Lambda_h^k)^{-1}}}. \tag{F.28}$$

If $n_0$ satisfy (F.28) for any $h \in [H]$, (F.27) indicates that the condition in (F.23) is satisfied and the estimation process is terminated. Therefore, we know that

$$n_0 \leq \left\lceil \log\left(\frac{3}{\max_{h \in [H]} \|\phi_h^k\|_{(\Lambda_h^k)^{-1}}}\right)\right\rceil.$$

Let $\iota' = \log(dHK/\delta)$, by the property of quantum multivariate mean estimation (Lemma 3.2 with $C = H$), we need at most

$$H \cdot \mathcal{O}\left(\sqrt{d}H\iota' \cdot \sum_{n=1}^{n_0+1} 2^n\right) \leq \mathcal{O}\left(\frac{\sqrt{d}H^2\iota'}{\max_{h \in [H]} \|\phi_h^k\|_{(\Lambda_h^k)^{-1}}}\right) \tag{F.29}$$

episodes to estimate features. By Lemma F.1 and the definitions of $\iota'$ and $\iota$, we have $\iota' \leq \mathcal{O}(\iota)$. Together with (F.29), we conclude the proof of Lemma F.3. □

### F.5 REGRESSION TARGET ESTIMATION

**Lemma F.4** (Estimate Regression Target). It holds with probability at least $1 - \delta/2$ that

$$\|y_h^k - (\phi_h^k)^\top \theta_h\|_2 \leq w_k$$

for all $(k, h) \in [K] \times [H]$. Meanwhile, for each $k \in [K]$, we need at most

$$\mathcal{O}\left(\frac{H^2\iota}{w_k}\right)$$

episodes to calculate $\{y_h^k\}_{h \in [H]}$. Here $\iota = \mathcal{O}\left(\log\left(\frac{dH \log(1+\frac{T^3}{d})}{\delta}\right)\right)$.

*Proof of Lemma F.4.* For any $h \in [H]$, invoking Lemma 3.2 with $d = 1$ and $C = H$, we know that we need at most

$$\mathcal{O}\left(\frac{H\iota}{w_k}\right)$$

episodes to calculate $y_h^k$. Taking summation over $h \in [H]$ concludes the proof of Lemma F.4. □

F.6 PROOF OF THEOREM 5.2

Our proof relies on the following value decomposition lemma.

**Lemma F.5** (Value Decomposition Lemma). Let $\pi = \{\pi_h\}_{h\in[H]}$ and $\widehat{\pi} = \{\widehat{\pi}_h\}_{h\in[H]}$ be any two policies and $\widehat{Q} = \{\widehat{Q}_h\}_{h\in[H]}$ be any estimated Q-functions. Moreover, we assume that the estimated value functions $\widehat{V} = \{\widehat{V}_h\}_{h\in[H]}$ satisfy $\widehat{V}_h(s) = \langle \widehat{Q}_h(\cdot\,|\,s), \widehat{\pi}_h(\cdot\,|\,s)\rangle_{\mathcal{A}}$ for all $(s,h) \in \mathcal{S}\times[H]$. Then we have

$$\widehat{V}_1(s_1) - V_1^\pi(s_1) = \sum_{h=1}^{H} \mathbb{E}_\pi \left[ \langle \widehat{Q}_h(s_h, \cdot), \widehat{\pi}_h(\cdot\,|\,s_h) - \pi_h(\cdot\,|\,s_h)\rangle_{\mathcal{A}} \right]$$
$$+ \sum_{h=1}^{H} \mathbb{E}_\pi \left[ \widehat{Q}_h(s_h, a_h) - R_h(s_h, a_h) - \mathbb{P}_h \widehat{V}_{h+1}(s_h, a_h) \right].$$

*Proof.* See Appendix B.1 of Cai et al. (2020) for a detailed proof. $\square$

*Proof of Theorem 5.2.* Under the event defined in Lemma F.2, we have $\theta_h \in \mathcal{C}_h^k$ for all $(k,h) \in [K]\times[H]$. Together with the optimistic planning (Algorithm 5), we have

$$V_1^*(s_1) \leq V_1^k, \quad \forall k \in [K]. \tag{F.30}$$

Hence, we further obtain that

$$V_1^*(s_1) - V_1^{\pi^k}(s_1) \leq V_1^k(s_1) - V_1^{\pi^k}(s_1)$$
$$= \sum_{h=1}^{H} \mathbb{E}_{\pi^k}[Q_h^k(s_h, a_h) - R_h(s_h, a_h) - \mathbb{P}_h V_{h+1}^k(s_h, a_h)], \tag{F.31}$$

where the inequality follows from (F.30) and the equality uses Lemma F.5. By the optimistic planning procedure (Algorithm 5), we have

$$\mathbb{E}_{\pi^k}[Q_h^k(s_h, a_h)] = \mathbb{E}_{\pi^k}\left[ R_h(s_h, a_h) + \phi_{V_{h+1}^k}(s, a)^\top \widehat{\theta}_h^k \right] = \mathbb{E}_{\pi^k}[R_h(s_h, a_h)] + (\phi_h^k)^\top \widehat{\theta}_h^k \tag{F.32}$$

where $\phi_{V_{h+1}^k}$ id defined in (2.5) and the last equality uses the definition of $\phi_h^k$ that $\phi_h^k = \mathbb{E}_{\pi^k}[\phi_{V_{h+1}^k}(s_h, a_h)]$. Meanwhile, by the same argument in (5.2), we have

$$\mathbb{E}_{\pi^k}[\mathbb{P}_h V_{h+1}^k(s_h, a_h)] = (\phi_h^k)^\top \theta_h. \tag{F.33}$$

Plugging (F.32) and (F.33) into (F.31) gives that

$$V_1^*(s_1) - V_1^{\pi^k}(s_1) \leq \sum_{h=1}^{H} [(\phi_h^k)^\top (\widehat{\theta}_h^k - \theta_h)]. \tag{F.34}$$

Recall that in the $k$-th phase, the learning process consists of the feature estimation and the regression target estimation. In the sequel, we establish the regret upper bound incurred by these two parts respectively.

**Feature Estimation Error:** Applying Cauchy-Schwarz inequality to (F.34) gives that

$$V_1^*(s_1) - V_1^{\pi^k}(s_1) \leq \sum_{h=1}^{H} \|\phi_h^k\|_{(\Lambda_h^k)^{-1}} \cdot \|\widehat{\theta}_h^k - \theta_h\|_{\Lambda_h^k}. \tag{F.35}$$

Under the event $\mathcal{E}^\dagger$ defined in Lemma F.2, we have

$$\|\widehat{\theta}_h^k - \theta_h\|_{\Lambda_h^k} \leq \|\widehat{\theta}_h^k - \bar{\theta}_h^k\|_{\Lambda_h^k} + \|\bar{\theta}_h^k - \theta_h\|_{\Lambda_h^k} \leq 2\beta_k. \tag{F.36}$$

for any $h \in [H]$. Together with (F.35), we further obtain that

$$V_1^*(s_1) - V_1^{\pi^k}(s_1) \leq 2\beta_k \sum_{h=1}^{H} \|\phi_h^k\|_{(\Lambda_h^k)^{-1}} \leq 2H\beta_k \cdot \max_{h\in[H]} \|\phi_h^k\|_{(\Lambda_h^k)^{-1}}. \tag{F.37}$$

By Lemma F.3, we know that we use at most

$$\mathcal{O}\Big(\frac{\sqrt{d}H^2\iota}{\max_{h\in[H]}\|\phi_h^k\|_{(\Lambda_h^k)^{-1}}}\Big) \tag{F.38}$$

episodes to estimate features. Combining (F.37) and (F.38), we obtain that the feature estimation error of phase $k$ is at most

$$\mathcal{O}(\sqrt{d}H^3\beta_k\cdot\iota). \tag{F.39}$$

**Regression Target Estimation Error:** By (F.34), we have

$$
\begin{aligned}
V_1^*(s_1) - V_1^{\pi^k}(s_1) &\leq \sum_{h=1}^{H}[(\phi_h^k)^\top(\widehat{\theta}_h^k - \theta_h)]\\
&= \sum_{h=1}^{H}[(\widehat{\phi}_h^k)^\top(\widehat{\theta}_h^k - \theta_h)] + \sum_{h=1}^{H}[(\phi_h^k - \widehat{\phi}_h^k)^\top(\widehat{\theta}_h^k - \theta_h)]\\
&\leq \sum_{h=1}^{H}\|\widehat{\phi}_h^k\|_{(\Lambda_h^k)^{-1}}\cdot\|\widehat{\theta}_h^k - \theta_h\|_{\Lambda_h^k} + \sum_{h=1}^{H}\|\phi_h^k - \widehat{\phi}_h^k\|_2\cdot\|\widehat{\theta}_h^k - \theta_h\|_2, \quad (F.40)
\end{aligned}
$$

where the last inequality uses the Cauchy-Schwarz inequality. Under the event $\mathcal{E}^\dagger$ defined in Lemma F.2, by the same derivation of (F.36), we have

$$\|\widehat{\theta}_h^k - \theta_h\|_{\Lambda_h^k} \leq \|\widehat{\theta}_h^k - \bar{\theta}_h^k\|_{\Lambda_h^k} + \|\bar{\theta}_h^k - \theta_h\|_{\Lambda_h^k} \leq 2\beta_k. \tag{F.41}$$

Meanwhile, by Lemma F.3 and (2.4), we have

$$\|\phi_h^k - \widehat{\phi}_h^k\|_2\cdot\|\widehat{\theta}_h^k - \theta_h\|_2 \leq 2w_k. \tag{F.42}$$

Plugging (F.41) and (F.42) into (F.40), we have

$$
\begin{aligned}
V_1^*(s_1) - V_1^{\pi^k}(s_1) &\leq 2\beta_k\sum_{h=1}^{H}\|\widehat{\phi}_h^k\|_{(\Lambda_h^k)^{-1}} + 2Hw_k\\
&\leq 2\beta_k Hw_k + 2Hw_k = 2Hw_k(1+\beta_k). \quad (F.43)
\end{aligned}
$$

By Lemma F.4, we know that we need at most

$$\mathcal{O}\Big(\frac{H^2\iota}{w_k}\Big) \tag{F.44}$$

episodes to estimate regression targets. Putting (F.43) and (F.44) together, we have that the regression target estimation error of phase $k$ is at most

$$\mathcal{O}\Big(H^3\beta_k\cdot\iota\Big). \tag{F.45}$$

Combining (F.39) and (F.45), we have that the regret incurred by the $k$-th phase is at most

$$\mathcal{O}(\sqrt{d}H^3\beta_k\cdot\iota) = \mathcal{O}(dH^3\sqrt{k}\cdot\iota), \tag{F.46}$$

where we uses that $\beta_k = 1 + 2\sqrt{dk}$. Hence, the total regret is upper bounded by

$$\text{Regret}(T) \leq \sum_{k=1}^{K}\mathcal{O}(dH^3\sqrt{k}\cdot\iota) \leq \mathcal{O}(dH^3K^{3/2}\cdot\iota) \leq \mathcal{O}\Big(d^{5/2}H^{9/2}\log^{3/2}\big(1+\frac{T^3}{d}\big)\cdot\iota\Big),$$

where the first inequality uses (F.46) and the last inequality follows from Lemma F.1. Therefore, we finish the proof of Theorem 5.2. $\qquad\square$

