# OpenReview forum: "Provably Efficient Exploration in Quantum Reinforcement Learning with Logarithmic Worst-Case Regret"
_ICLR.cc/2024/Conference — Submitted to ICLR 2024_

### Official Review · Reviewer_AteR · 2023-10-24

**Soundness:** 3 good
**Presentation:** 3 good
**Contribution:** 2 fair
**Rating:** 6
**Confidence:** 3

**Summary:**

This work studies quantum reinforcement learning, where quantum means that the classical reward and state transition feedback is replaced by quantum pure states (see Eq. (3.1) (3.2)). This paper studies both the general MDP and linear MDP, and it shows that they can achieve logarithmic regret performance, which breaks the classical square-root regret lower bound.

**Strengths:**

- The authors did a good job of presenting this work and comparing it to known literature.

**Weaknesses:**

- Lack of novelty. I am familiar with the work of Wan et al 2022 for quantum multi-armed bandits and quantum linear bandits. As the main theoretical tools for multi-armed bandits and linear bandits are considerably similar to RL and linear RL respectively, this paper can be regarded as an extension from Wan et al 2022 to quantum RL. Although the author pointed out one new challenge in Remark 5.1, I did not see enough novel contributions in this work.



---
Zongqi Wan, Zhijie Zhang, Tongyang Li, Jialin Zhang, and Xiaoming Sun. Quantum multi-armed
bandits and stochastic linear bandits enjoy logarithmic regrets. In To Appear in the Proceedings
of the 37th AAAI Conference on Artificial Intelligence, 2022. arXiv:2205.14988

**Questions:**

Although leaning toward a negative evaluation of this work for its lack of contribution, I think this quantum RL topic is interesting and would suggest that the authors look into challenging issues around this topic, e.g., regret lower bounds for quantum RL which is not studied in quantum bandits in Wan et al 2022.

If the authors think there are other nontrivial challenges (except for Remark 5.1) in this work than in Wan et al 2022, please take the chance of rebuttal to explain.

---

> ### Author Response · Authors · 2023-11-15
>
> We appreciate the thoughtful review and have carefully considered the concerns raised. We would like to address the primary issue of perceived lack of novelty by providing a detailed clarification of the unique contributions of our work.
>
> **Regarding lack of novelty.**  Our work acknowledges its relationship with [1], extensively discussed in the related work and main paper, particularly in Remark 5.1. While it is true that multi-armed bandits and linear bandits are special cases of tabular and linear mixture MDPs, respectively, learning MDPs with multi-step decision-making introduces substantial complexity beyond bandit scenarios. We highlight the technical difficulties and our novel contributions.
>
>
> **Algorithm design novelties:**
> - One can regard the bandits as the MDPs satisfying (i) the state space only contains a single dummy state $s_{\mathrm{dummy}}$; (ii) $H=1$; and (iii) the reward only depends on the action chosen by the learner, i.e., $r(s_{\mathrm{dummy}}, a) = r(a)$. In this case, the learner can repeatedly pull a particular arm $a$ to collect sufficient samples to estimate $r(s_{\mathrm{dummy}}, a)$. However, in online RL, the state will transit according to the transition kernel, preventing repeated arm pulls for the desired estimator. To address this, we introduce a novel adaptive lazy-updating rule, quantifying the uncertainty of the visiting state-action pair (covariance matix in linear setting) and updating the model infrequently through the doubling trick. **This algorithm design, absent in previous works on quantum bandits [1] and quantum RL with a generative model [2], connects online exploration in quantum RL with strategic lazy updating mechanisms, inspiring subsequent algorithm design in online quantum RL.**
>
>
> **Technical novelties:**
>  - In the **tabular** setting, a significant technical challenge is obtaining a high probability regret bound, as vanilla regret decomposition (e.g., the one used in UCBVI) leads to a martingale term of order $\mathcal{O}(H\sqrt{T})$. Though it is not the dominating term in the classical setting, it would become the dominating term in our setting. **We found that another regret decomposition (i.e., the one used in EULER) based on the expected Bellman error cleverly bypasses such martingale terms in the regret, thus achieving the desired regret bound.**
>  - In the **linear** setting, the algorithm requires a novel design to leverage the advantage of quantum mean estimation. Classical linear RL analysis heavily relies on martingale concentration, such as the well-known self-normalized concentration bound for vectors. However, in quantum RL, there is no direct counterpart to martingale analysis. Consequently, **we redesign the lazy updating frequency and propose an entirely new technique to estimate features for building the confidence set for the model.** Notably, previous works on quantum bandits [1] do not need to take this step since there is no unknown transition kernel in their setting. Moreover, estimating the features poses a **subtle technical challenge, as elaborated in Equation (5.5).** Our proposed algorithm (Algorithm 2), which features **binary search and quantum mean estimation**, successfully addresses this technical challenge. Meanwhile, **the quantum samples used in feature estimation approximately equal the number of quantum samples in each phase, eliminating extra regret for this additional feature estimation.** This algorithm design (Algortihm 2) and theoretical analysis are **entirely new** in the literature on (classical and quantum) RL theory. We sincerely hope the reviewer can reconsider the novelty of this part. We believe that **introducing a new algorithm design and novel techniques with interesting results is sufficient for the first theoretical study in online quantum RL (with function approximation).**
>
> **Significance of our results:**
> - Beyond our novelties in algorithm design and techniques, the paramount significance lies in the message that quantum computing can potentially provide a speedup for online sequential decision-making problems under the framework of MDPs. The logarithmic regret obtained is markedly different from classical $\sqrt{T}$ regret, providing a theoretical guarantee for the potential superiority of quantum RL over classical RL. As **the first rigorously theoretical work on online quantum RL**, our research is deemed significant, paving the way for future investigations in the realm of quantum RL.
>
>
> According to your concerns, we have made a thorough revision of our description of novelties. Please refer to the **introduction section, Remark 5.1, the proof sketch of Theorem 5.2, and Appendix B.2 for details, as highlighted in blue.**
>
> **Regarding regret lower bound.** Please see our general response for details.
>
>
> [1] Quantum multi-armed bandits and stochastic linear bandits enjoy logarithmic regrets.
>
> [2] Quantum algorithms for reinforcement learning with a generative model.

---

> > ### Comment · Reviewer_AteR · 2023-11-19
> >
> > The reviewer thanks the author for providing additional explanations about this work's novelty. The reviewer will take these into consideration in the later discussion with AC.

---

> > > ### Author Response · Authors · 2023-11-19
> > >
> > > Thanks for reading our response and recognizing the novelties of our work. Please let us know if you have any further questions. If your concern is addressed, we would appreciate if you would reconsider your score in light of our clarification. Thank you!

---

### Official Review · Reviewer_pp1B · 2023-10-29

**Soundness:** 3 good
**Presentation:** 3 good
**Contribution:** 3 good
**Rating:** 6
**Confidence:** 4

**Summary:**

This paper studies quantum RL, which provides sample complexity for both tabular MDP and linear mixture MDP, based on several quantum estimation oracles. Compared with previous literature, this paper provides an online exploration method for quantum RL.

**Strengths:**

- This paper is well written and easy to follow
- Study quantum RL is novel in the literature, with limited prior works
- The proposed online exploration paradigm is more practical than previous work.

**Weaknesses:**

- The discussion on sample complexity is not enough. For example, it would be better to discuss why the $\sqrt{T}$ factor is removed. Is that because of Lemma 3.1 and Lemma 3.2 such that the previous $\epsilon^{-2}$ sample complexity can be reduced to $\epsilon^{-1}$ sample complexity so that the exploration can be more aggressive?
- Besides the previous comment, I'm also looking for discussions about the lower bounds (or at least some conjectures). For example, if the dependency on $d, H$ within the lower bounds still match (Zhou et al., 2021) or not?

**Questions:**

Besides my concern about the weakness, I'm concerned about the cost of translating a classical RL task into a quantum-accessible RL task. Here are my questions
- Can one directly covert the observation in classical RL to a quantum-accessible RL? (e.g., changing the Atari games to quantum). If the quantum RL can be used in classical RL tasks, then how would the current $\log T$ bound break the classical $\sqrt{T}$ regret bound?
- If the current algorithm can only be used in quantum-accessible RL, and we cannot convert a classical RL task into quantum, then how will this algorithm contribute to real-world RL tasks?

---

> ### Author Response · Authors · 2023-11-15
>
> Thanks for your review and positive feedback. We will try to address your concerns in the following.
>
> **Q1:** The discussion on sample complexity is not enough. For example, it would be better to discuss why the $\sqrt{T}$ factor is removed. Is that because of Lemma 3.1 and Lemma 3.2 such that the previous $\epsilon^{-2}$ sample complexity can be reduced to $\epsilon^{-1}$ sample complexity so that the exploration can be more aggressive?
>
> **A1:** Yes, the logarithmic regret in our work is primarily attributed to the speedup of quantum mean estimation (Lemma 3.1 and Lemma 3.2). Simultaneously, the $\epsilon^{-1}$ sample complexity leads to more conservative exploration instead of aggressive, as we can make more accurate estimations. This conservatism is further reflected in the bonus term; our bonus term $1/n_h^k(s, a)$ is smaller than the classical $1/\sqrt{n_h^k(s, a)}$, resulting in a more cautious exploration strategy. We appreciate your valuable insights and comments, and we have incorporated these discussions into the introduction, highlighted in blue.
>
> Moreover, the strategic combination of the speedup in quantum mean estimations and exploration in online quantum RL is the central focus of our algorithm design and analysis. Please refer to our introduction section and other parts of the main paper for a detailed elaboration of this aspect.
>
> **Q2:** Besides the previous comment, I'm also looking for discussions about the lower bounds (or at least some conjectures). For example, if the dependency on $d, H$ within the lower bounds still match (Zhou et al., 2021) or not?
>
> **A2:** Please see our general response for details.
>
> **Q3:** Can one directly covert the observation in classical RL to a quantum-accessible RL? (e.g., changing the Atari games to quantum). If the quantum RL can be used in classical RL tasks, then how would the current $\log T$ bound break the classical $\sqrt{T}$ regret bound?
>
> **A3:** We can run classical RL on quantum computers as a special case. For instance, we can play classical Atari games on quantum computers when they become universal and have enough number of qubits to execute the programs. That is, quantum computers can simulate classical RL environments. Moreover, quantum accessible environments empower the agent to hold the quantum superpositions over states and actions, which encode the stochasity of the envrionments. In contrast, classical environment only returns random samples from the environment.
>
>
>
> **Q4:** If the current algorithm can only be used in quantum-accessible RL, and we cannot convert a classical RL task into quantum, then how will this algorithm contribute to real-world RL tasks?
>
> **A4:** Same as A3, we can regard a classical RL task as a quantum RL task but never uses superposition. The algorithm proposed in this paper is a quantum RL algorithm and can only be executed on quantum computers. In terms of the contribution to real-world RL tasks, due to the limitation of current quantum computers we have to focus on theoretical research of quantum RL. Nevertheless, our result is fundamental as we proved the first poly-log regret results for quantum RL, and we believe that our work will have real-world impact after the capability of quantum computers significantly grows.

---

### Official Review · Reviewer_KqvL · 2023-10-31

**Soundness:** 3 good
**Presentation:** 1 poor
**Contribution:** 3 good
**Rating:** 6
**Confidence:** 3

**Summary:**

This work studies a quantum RL problem, where the objective is to explore the episodic MDP with quantum access and learn the optimal policy while minimizing the regret over $T$ episodes. The authors propose the Quantum UCRL and Quantum UCRL-VTR algorithms for tabular MDP and linear mixture MDP settings respectively. Their analysis of the algorithms gives $O(\text{poly}(\log T))$ regret upper bounds.

**Strengths:**

- This work is one of the pioneering effort in studying quantum reinforcement learning with theoretical guarantees.
- This work incorporates quantum *multi-dimensional/multivariate* estimation subroutines into UCRL-based algorithms. The insights in such incorporations may of interest to the emerging quantum machine learning community.

**Weaknesses:**

- Quantum regret lower bound is not discussed in the paper
- The presentation is not clear -- some notations are used without clearly defined or explained
- A closely related work is missing from the literature review: Ganguly, Bhargav, et al. "Quantum Computing Provides Exponential Regret Improvement in Episodic Reinforcement Learning." arXiv preprint arXiv:2302.08617 (2023).
- This work does not have any empirical study of the proposed algorithms

**Questions:**

- Could the authors comment on why the binary oracle is not considered in the tabular setting? What is the main difficulty in generalizing the result of Lemma 3.1 to binary oracle?
- Is the regret lower bounds of the studied "quantum" exploration problems known? If not, could the authors comment on the difficulties of getting such lower bounds?

The above two points may be worth mentioning in a future direction section/paragraph.

- The introduction/description of the Quantum UCRL algorithm is not clear enough. Specifically, I could not find the $\bar{\varphi}_{h+1}, \mathcal{D}_h(s^k_h, a^k_h)$ notations appeared in Algorithm 4 being defined anywhere.
- If $\bar{\varphi}_{h+1}$ is as defined at the end of subsection 3.2, then it is a quantum state in superposition. How could Algorithm 4 line 9 update the counter according to the superposition? Please correct me if I missed anything.
- Why does Quantum UCRL divides the episodes into T/H phases while Quantum UCRL-VTR divides into K phases? How should the practitioners set the parameter K for Quantum UCRL-VTR?

I would love to see Algorithm 4 be presented in the main paper for the sake of clarity if the page limit allows.

- (minor wording issue)  The term "quantum state" is somehow ambiguous as the term "state" has its special meaning in RL problem.

---

> ### Author Response · Authors · 2023-11-15
>
> Thanks for your review and positive feedback. We will try to address your concerns in the following.
>
> **Q1:** Regret lower bound.
>
> **A1:** Please see our general response for details.
>
> **Q2**: The introduction/description of the Quantum UCRL algorithm is not clear enough. For example, the notation $\bar{\varphi}_{h+1}, \mathcal{D}_h(s_h^k, a_h^k)$ appeared in Algorithm 4 is not defined clearly.
>
> **A2**: We greatly appreciate the reviewer's suggestion to clarify the confusing notations/definitions in our paper. To be specific, the place $\bar{\varphi}\_{h+1}$ appeared in Algorithm 4 (i.e., Line 7 of Algorithm 4) is exactly the definition of $\bar{\varphi}\_{h+1}$. According to the context, $\bar{\varphi}\_{h+1}$ is **defined as** the output of the oracle $\bar{\mathcal{P}}\_h$ (defined by Eqn. (3.2)) queried on input $\left|s_h^k, a_h^k\right\rangle|0\rangle$. We should use $\ket{\bar{\varphi}\_{k, h+1}}$ to represent this output because it is a quantum superposition obtained in episode $k$. Similarly, the definition of $\bar{\varphi}\_{h+1}^{-1}$ is $\bar{\varphi}\_{h+1}^{-1} := \bar{\mathcal{P}}\_h^{-1} \left|s_h^k, a_h^k\right\rangle|0\rangle$ (we will also use $\ket{\bar{\varphi}\_{k,h+1}^{-1}}$ instead of $\bar{\varphi}\_{h+1}^{-1}$ in the revised version of the paper).
>
> The notation $\mathcal{D}_h(s_h^k, a_h^k)$ is defined at Line 2 of Algorithm 4 (the third bullet point). It is defined as the collection (i.e., the multiset) of all the quantum samples obtained for the state-action pair $(s_h^k, a_h^k)$ at step $h$. We will use multiset $\mathcal{D}_h(s_h^k, a_h^k)$ as input to call the quantum multi-dimensional amplitude estimation algorithm (Lemma 3.1).
>
> **Q3**: $\bar{\varphi}\_{h+1}$ is a quantum state in superposition. How could Algorithm 4 line 9 update the counter according to the superposition? Please correct me if I missed anything.
>
> **A3**: According to the definition, $\bar{\varphi}\_{h+1} = \bar{\mathcal{P}}\_{h} \ket{s_{h}^k, a_{h}^k} \ket{0}$ is indeed a quantum superposition, with which we cannot update the counter. Fortunately, $(s_h^k, a_h^k)$ is a classical state-action pair instead of a quantum superposition, and $\ket{s_h^k, a_h^k}$ is a standard basis vector of Hilbert space $\bar{S} \times \bar{A}$ (defined at the beginning of the second paragraph of Section 3.2) encoded by $(s_h^k, a_h^k)$. This is because $s_h^k$ is the output of Algorithm 3, which always outputs a classical state sampled from some distribution. $a_h^k$ is also a classical action since $a_h^k := \pi_h^k(s_h^k)$. Therefore, we can update the counter according to $(s_h^k, a_h^k)$. Moreover, this is a key reason why we need to design a complicated phase-based "sampling" procedure in Algorithm 4 to obtain valid quantum samples, which is a dramatic difference between quantum RL and classical RL.
>
> **Q4**: Why the binary oracle is not considered in the tabular setting? What is the main difficulty in generalizing the result of Lemma 3.1 to binary oracle?
>
> **A4**: We can definitely generalize the results of Lemma 3.1 to Lemma 3.2 with binary oracle in the tabular setting. This is done in the following way. For a fixed $(s, a)$ pair, we define the binary oracle $U_{s, a} : \ket{s, a} \ket{s'} \ket{0} \to \ket{s, a} \ket{\vec{1}[s']}$, where $\vec{1}[s']$ is an $S$-dimensional standard basis vector encoded by $s'$ (i.e., a one-hot vector with non-zero entry at the $s'$ position). Combined with the probability oracle $\bar{\mathcal{P}}\_h: \ket{s, a} \ket{0} \to \sum_{s'} \sqrt{\mathcal{P}\_h(s' \mid s, a)} \ket{s'}$, we can estimate the $S$-dimensional vector $(\mathcal{P}\_h(s_1 \mid s, a), \mathcal{P}\_h(s_2 \mid s, a), ..., \mathcal{P}\_h(s_{S} \mid s, a))^\top \in \mathbb{R}^S$ suppose $s'$ is enumerated from the state space $\{s_1, s_2, ..., s_{S}\}$ **using Lemma 3.2**. Since we hope the $l_1$ estimation error to be bouned by $\epsilon$, the sample complexity of such estimation is $\mathcal{O}(S \log (S/\delta) / \epsilon)$, the same as Lemma 3.1. Since the target vector $(\mathcal{P}\_h(s_1 \mid s, a), \mathcal{P}\_h(s_2 \mid s, a), ..., \mathcal{P}\_h(s_{S} \mid s, a))$ is actually a distribution over the state space, we can also encode this vector as the amplitude of a quantum superposition and use Lemma 3.1 to estimate the amplitude. *We choose Lemma 3.1 mainly because the quantum multi-dimensional amplitude estimation subroutine is conceptually simpler without requiring an additional binary oracle, so it is more efficient in implementation.*

---

> > ### Author Response · Authors · 2023-11-15
> >
> > **Q5**: Why does Quantum UCRL divide the episodes into T/H phases while Quantum UCRL-VTR divides into K phases? How should the practitioners set the parameter K for Quantum UCRL-VTR?
> >
> > **A5**: Both the Quantum UCRL and Quantum UCLR-VTR algorithms leverage the idea of lazy updating: we divide the total $T$ episodes into $K (K \leq T)$ phases (each phase contains some contiguous episodes), and update the estimation of the parameter **only once in a single phase**. The purpose of lazy updating is to use quantum samples efficiently, since any quantum sample can be used only once (after then it collapses). The tabular setting is easier, in which we can use schedule the length of each phase to $H$ equally. The linear setting is more complicated, as we need to schedule the length of each phase **on the fly**: we decide the length of next phase at the end of last phase (the length of the $k$-th phase is $\tilde{\mathcal{O}}(\sqrt{d}H^2 / w_k)$ where $w\_k := \max\_{h \in[H]}||\widehat{\phi}\_h^k||\_{\left(\Lambda_h^k\right)^{-1}}$ according to Line 6 of Algorithm 1). It is guaranteed that the total number of phases will be bouned by $\mathcal{O}(dH \log T)$ according to Lemma F.1. Therefore, we do not need to choose the parameter $K$ at the beginning of Quantum UCRL-VTR. It suffices to follow the decision rule of the next phase length of Quantum UCRL-VTR. **We have added a proof sketch of Theorem 5.2 (see the second paragraph below Theorem 5.2, highlighted in blue), where we clarify the size of the phase number $K$ and the number of episodes in the $k$-th phase. Thank you for your question!**
> >
> > **Q6**: The term "quantum state" is somehow ambiguous as the term "state" has its special meaning in RL problem.
> >
> > **A6**: Quantum state is a standard definition in quantum computing. It is a coincidence that quantum computing literature and reinforcement learning literature both use the word "state", though for different meanings. In our paper, classical states are represented by a single character $s$, while quantum states are denoted in the ket form $\ket{s}$. This distinction should help avoid confusion. To address the reviewer's concern, we will further emphasize whether we are referring to classical states (in the context of reinforcement learning) or quantum states (pertaining to quantum computing and quantum RL) throughout our paper. Thank you for your helpful suggestion.
> >
> >
> > **Q7**: This work does not have any empirical study of the proposed algorithms.
> >
> > **A7**: While we appreciate the suggestion for an experimental study, our primary focus is on advancing the theoretical understanding of quantum reinforcement learning (RL). Given current limitations in quantum computing capabilities, our strategic emphasis on theory aims to lay a foundational framework. We believe that this theoretical groundwork will inspire both theoretical and empirical advancements in quantum RL as quantum computing technology evolves.

---

> > > ### Author Response · Authors · 2023-11-15
> > >
> > > **Q8**: Comparison to Ganguly, Bhargav, et al. "Quantum Computing Provides Exponential Regret Improvement in Episodic Reinforcement Learning." arXiv preprint arXiv:2302.08617 (2023).
> > >
> > > **A8**: Thank you for pointing out this work. Importantly, we want to emphasize that this is a *concurrent and independent* work. Despite this, we remain open to discussing the distinctions from it. The concurrent and independent work  (Ganguly et al. 2023) also focuses on quantum RL and establishes logarithmic regret for **tabular** MDPs. The main differences between (Ganguly et al. 2023) and our work includes:
> > >
> > > - The problem settings are different. (Ganguly et al. 2023) assumes an MDP proceeding classically, with an agent providing a certain action $a_h$ at time step $h$, getting reward $r_h$, and stepping to next state $s_{h+1}$ from $s_h$. Here $(s_h, a_h, r_h, s_{h+1})$ are collected as classical information. For each step, the agent is assumed to observe $S$ additional quantum samples returned by $S$ unitaries: $U_s \ket{0} = \sqrt{1-\mathcal{P}\_h(s_{h+1} = s \mid s_h,a_h)}\ket{0} + \sqrt{\mathcal{P}\_h(s_{h+1} = s\mid s_h,a_h)}\ket{1}, \forall s \in \mathcal{S}$, which encode the probability of making transition to $s_{h+1}$ by taking action $a_h$ at state $s_h$ in order to estimate the distribution $\mathcal{P}\_h(s_{h+1} = \cdot \mid s_h,a_h) \in \mathbb{R}^S$. In contrast, our work considers a quantum-accessible environment. In our work, the agent is allowed to take a quantum policy as (3.4), which is stochastic and is naturally evaluated by unitaries. The transition and reward are also given by quantum unitaries as (3.2) and (3.3). The whole episode of MDP is quantized and it is natural to quantum computers. Due the the different models studied by (Ganguly et al. 2023) and us, our regret bound is worse than their regret bound by a factor of $H$. We want to emphasize that our analysis immediately implies the same regret bound as theirs under their setting.
> > > - More importantly, our work also considers linear mixture MDPs, which are not considered by (Ganguly et al. 2023). This setting is significantly more challenging than the tabular setting. To establish the logarithmic regret for linear mixture MDPs, we develop new algorithm designs and theoretical analysis, as detailed in the "Challenges and Technical Overview'' part of the introduction section, Section 5, and Appendix B.2.
> > >
> > > The comparison between our work and the mentioned work has been added in Appendix A.

---

> ### Comment · Reviewer_KqvL · 2023-11-22
>
> Thanks for the authors' detailed response to my questions. I would encourage the authors to add into the paper the discussions about updating sample counters (as reviewer XG1J also has a similar concern) and generalizing binary oracle to tabular setting to make the paper more complete.

---

> > ### Author Response · Authors · 2023-11-22
> >
> > Thank you for your ongoing effort and support. We have incorporated these discussions into Remark C.4 and Remark D.1 in our revised manuscript.

---

### Official Review · Reviewer_XG1J · 2023-11-22

**Soundness:** 3 good
**Presentation:** 3 good
**Contribution:** 3 good
**Rating:** 6
**Confidence:** 3

**Summary:**

This paper studies the online exploration problem in reinforcement learning. Specifically, two RL settings are considered: tabular Markov decision processes (MDPs) and linear mixture MDPs; and the goal is to learn the policy that minimizes regret. To achieve this, the authors propose two algorithms that adapt existing RL algorithms by using tools from quantum computing to get performance gain.

**Strengths:**

**The following are the strengths of the paper:**
1. Adapting recent tools from quantum computing to improve the performance of RL algorithms is a challenging and interesting contribution.

2. The authors consider two RL settings -- tabular MDPs and linear mixture MDPs; and propose algorithms (Quantum UCRL and Quantum UCRL-VTR) with logarithmic regret (in terms of episodes) for both problems due to quantum speedup.

**Weaknesses:**

**The following are the key weaknesses of the paper:**
1. Motivating examples: It is unclear if the assumptions (access to quantum oracles and their inverse, quantum state) made in the paper are practical or not. Adding a few motivating examples where these assumptions (will) hold will make the contribution even more significant.

2. The doubling trick to design lazy-updating algorithms with quantum estimators is already used in existing work (e.g., Wan et al., 2022), so saying this is a  novel technique proposed in the paper is an overclaim (Last paragraph on Page 2). However, I agree adapting this idea to MDPs is not that straightforward.

3. Since the learner does not observe the next state, it is unclear how the number of quantum samples ($n_h(s, a)$) is tracked by the learner. It is important as tracking $n_h(s, a)$ is needed to update the estimate of the transition kernel. Overall, adding a detailed explanation of how quantum computing tools are used will make it easier to understand the contributions.

**Questions:**

Please address the above weaknesses. I have a few more questions/comments:
1. Page 6, paragraph before 'Lazy updating via doubling trick': Is there any connection between phase length (H) and episode horizon (H)?

2. The quantum oracle for reward function is not used as it is assumed to be known for the problems considered in the paper. Is this right?

Minor comment:
 If possible, authors can add a few experiments using the Python library QisKit. It will make the paper stronger.

I am open to changing my score based on the authors' responses.

---

> ### Author Response · Authors · 2023-11-22
>
> Thanks for your review. We would like to address your concerns as follows.
>
> **Q1**: Motivating examples: It is unclear if the assumptions (access to quantum oracles and their inverse, quantum state) made in the paper are practical or not. Adding a few motivating examples where these assumptions (will) hold will make the contribution even more significant.
>
> **A1**: Sure, thanks for the suggestion.
>
> Intuitively, **as long as a classical RL task can be written as a computer program with source code, the quantum oracles we assumed can be instantiated.** This is because classical programs with source code can in principle be written as a Boolean circuit whose output follows the distribution p(·|s, a), and it is known that any classical circuit with N logic gates can be converted to a quantum circuit consisting of O(N) logic gates that can compute on any quantum superposition of inputs (see for instance Section 1.5.1 of [1]). For instance, Atari games can be played with our quantum oracles when quantum computers become universal and have enough number of qubits to execute the programs. On the other hand, we also need to point out that **the quantum oracle assumption does not apply to every MDP.** For instance, we cannot instantiate a quantum oracle for the position of robot arms because robot arms can only exist in the classical world, and their positions cannot be queried in superposition. This is further clarified in the revised version of the paper (see the blue lines between Eq. (3.3) and (3.4)).
>
> We would also like to mention that **our assumptions about quantum oracles (i.e, transition oracle, reward oracle, and policy oracle) are standard** in the literature of quantum RL and quantum computing. For instance, Ref. [2] assumes access to oracle $O\_x:|0\rangle \rightarrow \sum\_{\omega \in \Omega_x} \sqrt{P\_x(\omega)}|\omega\rangle|y^x(\omega)\rangle$ in equation (5), where $y^x:\Omega \rightarrow \mathbb{R}$ is the random reward associated with arm $x$ of multi-armed bandit. This is similar to our transition oracle $\bar{P}\_h$. In the multi-armed bandit problems, the reward of pulling arm $x$ is $y^x(\omega)$ with probability $P_x(\omega)$. Similarly, in the MDP problems, the agent is transferred to $s_{h+1}$ from $s_h$ by doing action $a_h$ with probability $P\_h(s_{h+1}|s_h,a_h)$, and we encode this transition as a probability oracle $\bar{P}\_h:|s_h,a_h\rangle|0\rangle\rightarrow|s_h,a_h\rangle\otimes\sum_{s_{h+1}}\sqrt{P_h(s_{h+1}|s_h,a_h)}|s_{h+1}\rangle$. Ref. [3] discusses quantum-accessible environments in section 2.3 and it defines transition oracle (8), reward oracle (9), and quantum evaluation of a policy (10) in definition 2.5 and 2.6. These definitions are completely the same as ours. Besides, Ref. [4] defines a quantum generative model of an MDP in definition 3, which is similar and of the same principle as our transition oracle $\bar{P}_h$.
>
> In addition, assuming the inverse of an oracle is natural and standard in quantum computing. Every oracle in quantum computing is a unitary and thus naturally reversible. After implementing a quantum oracle as a quantum circuit (which consists of only a series of basic quantum gates), the inverse can be implemented by directly implementing the inverse of every quantum gate in inverted order.
> For quantum states, we only assume access to initial state $|0\rangle^{\otimes n}$ and prepare other quantum states by basic quantum gates and our assumed oracles.
>
> **Q2**: The doubling trick to design lazy-updating algorithms with quantum estimators is already used in existing work (e.g., Wan et al., 2022), so saying this is a novel technique proposed in the paper is an overclaim (Last paragraph on Page 2). However, I agree adapting this idea to MDPs is not that straightforward.
>
> **A2**: We acknowledge that the algorithmic design of lazy-updating (e.g., the doubling trick) has been explored in the literature of classical online RL, and we have discussed these in our paper. We also appreciate that the reviewer agrees that adapting this idea to MDPs is not that straightforward.  In our paper, we have detailed the distinctions between quantum bandits and quantum RL in Appendix B.2. For the reviewer's convenience, we provide the details below.

---

> ### Author Response · Authors · 2023-11-22
>
> **A2 (continued):**
>
> - One can regard the bandits as the MDPs satisfying (i) the state space only contains a single dummy state $s_{\mathrm{dummy}}$; (ii) $H=1$; and (iii) the reward only depends on the action chosen by the learner, i.e., $r(s_{\mathrm{dummy}}, a) = r(a)$. In this case, the learner can repeatedly pull a particular arm $a$ to collect sufficient samples to estimate $r(s_{\mathrm{dummy}}, a)$. However, in online RL, the state will transit according to the transition kernel, preventing repeated arm pulls for the desired estimator. To address this, we introduce a novel adaptive lazy-updating rule, quantifying the uncertainty of the visiting state-action pair (covariance matrix in linear setting) and updating the model infrequently through the doubling trick. **This algorithm design, absent in previous work on quantum RL with a generative model [4], connects online exploration in quantum RL with strategic lazy updating mechanisms, inspiring subsequent algorithm design in online quantum RL.**
>
> Since the algorithmic design of the doubling trick is ubiquitous, **the term 'novel' in our context denotes a nontrivial adaptation of such designs to our model, specifically tailored for quantum-accessible tabular MDPs and linear mixture MDPs.** In response to the reviewer's concern, we have revised our paper to highlight that the primary contribution in this aspect lies in establishing a connection between online exploration in quantum RL and strategic lazy updating mechanisms
>
>
> Regarding technical novelty, we want to bring our linear mixture MDP part to the reviewer's attention. In the **linear** setting, the algorithm requires a novel design to leverage the advantage of quantum mean estimation. Classical linear RL analysis heavily relies on martingale concentration, such as the well-known self-normalized concentration bound for vectors. However, in quantum RL, there is no direct counterpart to martingale analysis. Consequently, **we redesign the lazy updating frequency and propose an entirely new technique to estimate features for building the confidence set for the model.** Notably, previous works on quantum bandits [2] do not need to take this step since there is no unknown transition kernel in their setting. Moreover, estimating the features poses a **subtle technical challenge, as elaborated in Equation (5.5).** Our proposed algorithm (Algorithm 2), which features **binary search and quantum mean estimation**, successfully addresses this technical challenge. Meanwhile, **the quantum samples used in feature estimation approximately equal the number of quantum samples in each phase, eliminating extra regret for this additional feature estimation.** This algorithm design (Algorithm 2) and theoretical analysis are **entirely new** in the literature on (classical and quantum) RL theory. Please refer to the **introduction section, Remark 5.1, the proof sketch of Theorem 5.2, and Appendix B.2 for more details, as highlighted in blue.**
>
>
>
> **Q3**: Since the learner does not observe the next state, it is unclear how the number of quantum samples ($n_h(s,a)$) is tracked by the learner. It is important as tracking $n_h(s,a)$ is needed to update the estimate of the transition kernel.
>
> **A3**: This is a great question. First, we would like to clarify that tracking the number of quantum samples $n_h(s, a)$ does not require the agent to know the next state, whether it is a classical state or a quantum state. Tracking $n_h(s, a)$ only requires knowing the classical state $(s, a)$ at step $h$, which is achieved by Algorithm 3 (CSQA) in Appendix C. This algorithm returns a classical state $(s, a)$ at step $h$ according to the given roll-out policy. When we use $(s, a)$ to query the transition oracle at step $h$, it is equivalent to apply $\bar{\mathcal{P}}_h$ on the input state $\ket{s, a} \ket{0}$, since $\ket{s, a}$ denotes the quantum representation of $(s, a)$ in the Hilbert space of quantum superpositions. A quantum sample is returned after the query, so we can increase the counter $n_h(s, a)$ by 1 since we gain a new independent quantum sample $\bar{\mathcal{P}}_h$ or its inverse queried on $\ket{s, a} \ket{0}$. We appreciate your valuable question, and to address this, we have incorporated a discussion in Remark C.4 in our revised manuscript, as highlighted in blue.
>
> **Q4**: Page 6, paragraph before 'Lazy updating via doubling trick': Is there any connection between phase length (H) and episode horizon (H)?
>
> **A4**: Yes, the phase length should be equal to the episode horizon. This is because Algorithm 3 (CSQA) only returns one classical sample at a given step $h$ according to a given roll-out policy in an episode, which means only one quantum sample is obtained in a single episode. Therefore, we need at least $H$ episodes in a phase to obtain a quantum sample of the roll-in policy at each step $h \in [H]$, so the model estimation can be updated with these quantum samples according to the lazy-updating scheme.

---

> > ### Author Response · Authors · 2023-11-22
> >
> > **Q5:** The quantum oracle for reward function is not used as it is assumed to be known for the problems considered in the paper. Is this right?
> >
> > **A5:** Yes, our quantum RL algorithms do not utilize the quantum oracle for the reward function under the assumption that the reward function is known. This assumption is mild and widely adopted in the RL theory literature.  It serves to simplify the presentation and theoretical analysis, yet retains its significance by addressing the most challenging aspect of RL --- learning unknown transition kernels. Furthermore, our results extend seamlessly to the unknown reward setting. In this scenario, we utilize quantum estimation oracles (Lemma 3.2) to estimate the unknown reward function. Notably, in the unknown reward setting, our final regret bounds remain consistent with those in the known reward setting (our current results). This alignment arises from the fact that the additional reward estimation error, roughly $1/n_h^k(s, a)$, is smaller than the transition estimation error, roughly $S/n_h^k(s,a)$. This reaffirms the notion that learning transition kernels poses a greater difficulty than learning the reward function, thereby lending support to the rationality of the known reward assumption.
> >
> >
> > **Q6:** Minor comment: If possible, authors can add a few experiments using the Python library QisKit. It will make the paper stronger.
> >
> > **A6:** Thank you for your suggestions. We want to emphasize that our work places a primary emphasis on the theoretical aspects of online quantum RL, with a central goal of providing the first logarithmic regret. We firmly believe that our theoretical results stand as a significant contribution in their own right and merit acceptance. Experiments are though interesting questions and can be left for future works.
> >
> >
> > ---
> >
> > As the discussion period is drawing to a close, we are unsure about the possibility of continuing our discussion to fully address your concerns. Nevertheless, we sincerely hope that the reviewer will carefully consider our responses, including the general response posted earlier, along with our revised paper. We firmly believe that the revisions and clarifications made enhance the value of our work and justify a reconsideration of its score. Your attention to this matter is greatly appreciated.
> >
> > ---
> > **References:**
> >
> > [1] Nielsen, Michael A., and Chuang, Isaac L.  Quantum Computation and Quantum Information. Cambridge University Press, 2010.
> >
> > [2] Wan, Zongqi, et al. "Quantum multi-armed bandits and stochastic linear bandits enjoy logarithmic regrets." Proceedings of the AAAI Conference on Artificial Intelligence. Vol. 37. No. 8. 2023.
> >
> > [3] Jerbi, Sofiene, et al. "Quantum policy gradient algorithms." arXiv preprint arXiv:2212.09328 (2022).
> >
> > [4] Wang, Daochen, et al. "Quantum algorithms for reinforcement learning with a generative model." International Conference on Machine Learning. PMLR, 2021.

---

### Author Response · Authors · 2023-11-15
**General Reponse: Discussions on regret lower bound and sample complexity.**

We appreciate the inquiries from all reviewers regarding the regret lower bound. Our main focus is to thoroughly address these concerns in this comprehensive response. In brief, a connection exists between sample complexity and regret bound, enabling us to establish a regret lower bound derived from the existing sample complexity lower bound. Subsequently, we will delve into a discussion about the gap between our upper bounds and lower bounds. Finally, we will offer clarification on our research focus.

**Relationship between regret guarantee and sample complexity:**
- **A regret upper bound guarantee implies a sample complexity upper bound.** If an algorithm executes policies $\\{\pi_i\\}\_{i=1}^T$ and incurs a regret $\mathrm{Reg}(T)$ in $T$ episodes, the algorithm can output a policy $\bar{\pi} = \mathrm{Uniform}(\\{\pi_i\\}_{i=1}^T)$ (uniformly select a policy). The gap between $\bar{\pi}$ and the optimal policy $\pi^*$ is $\mathrm{Reg}(T)/T$. Solving the inequality $\mathrm{Reg}(T)/T \le \epsilon$ gives the sample complexity of the algorithm. For example, a classical $\sqrt{T}$-regret upper bound implies an $\epsilon^{-2}$ sample complexity; our $\log T$-regret upper bound implies an $\epsilon^{-1}$ sample complexity.
- **A sample complexity lower bound implies a regret lower bound guarantee.** Assuming $g(\epsilon)$ is a known sample complexity lower bound, for any algorithm with regret $\mathrm{Reg}(T)$, it implies a sample complexity $f(\epsilon)$ as discussed above. Solving the inequality $f(\epsilon) \ge g(\epsilon)$ yields a regret lower bound. For example, an $\epsilon^{-2}$ sample complexity lower bound implies a $\Omega(\sqrt{T})$ regret lower bound; an $\epsilon^{-1}$ sample complexity lower bound implies a $\Omega(1)$ lower bound.

**Regret lower bound.** Since the MDP with a generative model is a simpler setting than the online quantum RL considered by us, the sample complexity lower bound $\Omega(S\sqrt{A}H^{1.5}/\epsilon)$ in the previous work [1] immediately implies the sample complexity lower bound for online quantum RL. As discussed above, this $\Omega(S\sqrt{A}H^{1.5}/\epsilon)$ sample complexity lower bound further implies an $\Omega(S\sqrt{A}H^{1.5})$ regret lower bound. Regarding the lower bound for linear mixture MDPs, we can regard the hard instance constructed in [1] as a linear mixture MDP with $d = S^2A$ (see also our discussions below Equation (2.5)). Hence, by the same proof in [1], we can obtain an $\Omega(\sqrt{dH^3}/\epsilon)$ sample complexity lower bound and an $\Omega(\sqrt{dH^3})$ regret lower bound for linear mixture MDPs.


**Further discussions on the gap between upper bounds and lower bounds.** Ignoring the logarithmic terms (in $T$ or $\epsilon$), our regret (sample complexity) can be improved in terms of $(S, A, H)$ ($(d, H)$). We conjecture that all these dependencies can be enhanced. One potential approach is using the variance reduction technique, as in classical online RL [2, 3]. However, we currently lack an understanding of how to apply this in the context of online quantum RL, which is far more sophisticated than classical counterpart mentioned in the paper. Moreover, deriving the minimax optimal sample complexity in MDPs with a generative model remains an open question, and addressing the sample complexity problem in this simpler setting seems more feasible and may provide more insights into obtaining sharper bounds in online quantum RL. As an initial exploration of online quantum RL, we leave achieving the tighter or even minimax optimal regret bound as future work.

**Clarifying our research focus.** Finally, we want to emphasize that the focus of our work is **making the first attempt to study online quantum RL and deriving the logarithmic regret bound for it, under both tabular and linear settings.** Our results (logarithmic regret or $\epsilon^{-1}$ sample complexity) have demonstrated the potential superiority of online quantum RL compared with classical RL. We believe this work provides a rigorous framework for future study, and the statistical limit of online quantum RL will be settled one day.


[1] Quantum Algorithms for Reinforcement Learning with a Generative Model. Daochen Wang, Aarthi Sundaram, Robin Kothari, Ashish Kapoor, Martin Roetteler.

[2]  Minimax regret bounds for reinforcement learning. Mohammad Gheshlaghi Azar, Ian Osband, and Remi Munos.

[3] Nearly minimax optimal reinforcement learning for linear mixture markov decision processes. Dongruo Zhou, Quanquan Gu, and Csaba Szepesvari.

---

> ### Comment · Reviewer_AteR · 2023-11-19
>
> The reviewer thanks the authors for providing the regret lower bounds, converted from sample complexity lower bounds. However, in sequential decision-making, the number of decision rounds $T$ is usually the largest factor. As $T$ does not appear in the given regret lower bounds, it is hard to use the lower bound to access the upper bounds provide in this paper.
>
> BTW, can the authors further explain how one could obtain a regret lower bound from $f(\epsilon)>g(\epsilon)$ in **A sample complexity upper bound implies a regret lower bound guarantee**? and it would be great if the author could provide some references for this statement.

---

> ### Author Response · Authors · 2023-11-19
>
> **A1:** Thank you for your question. According to our established regret lower bound, we can claim that our regret is **optimal in $T$ up to logarithmic terms**. While this assertion may seem trivial given that our regret bounds only have a logarithmic dependency in $T$, it is crucial to highlight that the inclusion of an additional logarithmic dependency in the upper bound is a common feature in existing RL theory literature. For instance, the regret lower bound for classical tabular MDPs is ${\Omega}(\sqrt{SAH^2T})$ [1, 2]. The regret upper bound for UCBVI [3] is $\mathcal{O}(\sqrt{SAH^2T} \cdot \mathrm{log}(T))$, which is considered nearly minimax optimal. Here, minimax optimal means that the upper bound matches the lower bound, **ignoring logarithmic terms**. We also want to emphasize that **in both classical bandits/RL and quantum bandits/RL, establishing a regret lower bound with logarithmic dependency is widely open. Given this context, it becomes particularly challenging to foresee the establishment of a $\Omega(\sqrt{S^2AH^3} \cdot \log T)$ or $\Omega(\sqrt{dH^3} \cdot \log T)$ regret lower bound.**
>
> **To demonstrate the sharpness of our results, one can also examine the comparison between the sample complexity upper bound and the sample complexity lower bound** to gain insights into the sharpness of the $\epsilon$ dependency, which corresponds to the dependency on $T$ in some sense (both quantify the number of episodes). Ignoring other problem parameters such as $S, A, d, H$, our regret upper bound implies a sample complexity upper bound $\mathcal{O}(\frac{1}{\epsilon}\cdot \log(\frac{1}{\epsilon}))$, and the sample complexity lower bound is $\Omega(\frac{1}{\epsilon})$. Hence, we can conclude that our sample complexity upper bound is **optimal in $\epsilon$ up to logarithmic factors**.
>
> ----
>
> **A2:** There is a typo in this statement. The correct statement is that **a sample complexity lower bound implies a regret lower bound guarantee**. Thank you for pointing this out, and we have revised the original response and paper accordingly. We also want to emphasize that our previous reduction statement argument still holds. To illustrate our reduction more, let us delve into the following specific scenario. Suppose there is a hard instance where every algorithm necessitates $C/\epsilon$ (resp. $C/\epsilon^2$) sample complexity to obtain the $\epsilon$ optimal policy, where $C$ involves certain problem parameters (e.g., $S, A, H, d$) and sufficiently large constants. Now, assume the existence of an algorithm that achieves regret $o(C)$ (resp. $o(\sqrt{CT})$). This implies an $o(C/\epsilon)$ (resp. $o(C/\epsilon^2)$) sample complexity, which contradicts the sample complexity lower bound. Hence, we can conclude that in this hard instance, every algorithm incurs a regret at least $\Omega(C)$ (resp. $\Omega(\sqrt{CT})$). **Intuitively,** regret appears to be a more "challenging" notion than sample complexity, as each regret upper bound has the potential to imply a sample complexity upper bound. Consequently, establishing a regret lower bound is intuitively considered to be a relatively "easy" task compared to the sample complexity lower bound.
>
> Regarding the reference, given the elementary nature of this result, we do not find the same form of statement in recent papers or books. However, we also do not want to claim this reduction as our new result and contribution. Moreover, we find that [2] shares a similar spirit with our reduction arguments. They provide (i) the reduction from the regret upper bound to the sample complexity upper bound; and (ii) an intuitive regret lower bound proof. In their lower bound proof, ignoring problem parameters, identifying the underlying MDP requires at least $\Omega(1/\epsilon^2)$ samples, and each will incur at least $\epsilon$ regret. Regarding $\Omega(1/\epsilon^2)$ as $T$, the final is at least $\Omega(1/\epsilon^2) \cdot \epsilon = \Omega(1/\epsilon) = \Omega(\sqrt{T})$ classical regret. Combining this proof paradigm and the sample complexity lower bound analysis in quantum RL with a generative model [5], identifying the underlying MDP requires $\Omega(1/\epsilon)$ quantum samples, and each will incur at least $\epsilon$ regret, further indicating the $\Omega(1)$ regret bound as desired. Notably, the core of this proof also implicitly converts the sample complexity lower bound to the regret lower bound. Despite slight differences from the proof paradigm in [2], we believe that our reduction statement is self-consistent and rigorous.
>
> ---
>
> In response to your inquiries, we have made revisions to **Appendix B.1** to provide clarification on these issues. We hope these revisions adequately address your concerns.

---

> > ### Author Response · Authors · 2023-11-19
> >
> > **References:**
> >
> > [1] Thomas Jaksch, Ronald Ortner, and Peter Auer. Near-optimal regret bounds for reinforcement learning. Journal of Machine Learning Research, 11:1563–1600, 2010.
> >
> > [2] Chi Jin, Zeyuan Allen-Zhu, Sebastien Bubeck, and Michael I. Jordan. Is Q-learning provably efficient? Advances in Neural Information Processing systems, 31, 2018.
> >
> > [3] Mohammad Gheshlaghi Azar, Ian Osband, and R´emi Munos. Minimax regret bounds for reinforcement learning. In International Conference on Machine Learning, pp. 263–272. PMLR, 2017.
> >
> > [4] Arjan Cornelissen, Yassine Hamoudi, and Sofiene Jerbi. Near-optimal quantum algorithms for multivariate mean estimation. In Proceedings of the 54th Annual ACM SIGACT Symposium on Theory of Computing, pp. 33–43, 2022.
> >
> > [5] Daochen Wang, Aarthi Sundaram, Robin Kothari, Ashish Kapoor, and Martin Roetteler. Quantum algorithms for reinforcement learning with a generative model. In International Conference on Machine Learning, pp. 10916–10926. PMLR, 2021a.

---

### Author Response · Authors · 2023-11-15
**Summary of Our Revision**

Based on the valuable feedback from all reviewers, we have uploaded a revised version of our paper to address the concerns. The key modifications are summarized below:

- We have added a comparison with a concurrent and independent work in Appendix A.
- In Appendix B.1, we have added a discussion on the regret lower bound and sample complexity.
- To provide a clearer exposition of our contributions, we have made revisions to the "Challenges and Technical Overview" section in the introduction. Additionally, we have refined Remark 5.1, included a proof sketch of Theorem 5.2, and incorporated more detailed discussions on our novel contributions in Appendix B.2.
- Various minor modifications have been implemented based on the suggestions from the reviewers.

All changes in this revision are highlighted in blue. We hope that these revisions effectively address the concerns raised by the reviewers. Should there be any additional questions from the reviewers, please do not hesitate to inform us, and we will be delighted to address them.

---

### Meta-Review · Area_Chair_pGQ8 · 2023-12-05

**Metareview:**

Due to the borderline and mixed reviews, I've read the paper myself.

This paper studies quantum RL, where the agent interacts with the environment by accessing quantum oracles. It proposed quantum variants of the classic UCRL and value target regression (VTR) algorithm that achieves log(T) regret in tabular and linear mixture MDPs respectively. The proposed algorithm, in particular, make use of a recently developed quantum mean estimation procedure to achieve the speedup.

Overall, I don't think the paper has done a good job motivating why quantum RL is a meaningful study at this point in time, where either quantum computing or RL still have many key challenges of their own to be solved in order to be of practical relevance. Judging it as a pure theory paper, the theoretical contribution seems mediocre, as most of the analysis remains the same as UCLR and VTR once you substitute the faster mean estimation rate from the quantum oracle.

The description of the problem setting is also ambiguous, at least to readers without prior quantum backgrounds (that is most readers of ICLR papers). It's not clear whether the paper is studying the same setting as online RL except that quantum tools are introduced as computational oracles, or if it's studying a fundamentally different problem.

My recommendation to the authors therefore is to describe more clearly the setting studied in the paper and the role of quantum tools in such settings.

SAC note: I generally agree with the AC's assessment. To add about the point of unclear setup: the revision mentioned that

> As long as a classical RL task can be written as a computer program with source code, we can perform our quantum RL algorithm with these quantum oracles.

This implies that we are solving a planning problem (model is given and known) using quantum tools as computational oracles. However, if an MDP is already coded, we have a generative model and are technically in the generative planning setting, which is a stronger protocol than online RL. The paper distinguishes itself from prior works on quantum RL for planning with a generative model (e.g., Wang et al 2021a) based on the online RL setting, but failed to provide a reasonable setup where quantum online RL is possible but generative planning is not.

**Justification For Why Not Higher Score:**

It can be higher score, but definitely one of the least impactful in my batch.

**Justification For Why Not Lower Score:**

NA

---

### Decision · Program_Chairs · 2024-01-16

Reject